# Auditor Fairness Evaluation via Learning Latent Assessment Models from Elicited Human Feedback - Rebuttal (ver.2)

## Abstract

Algorithmic fairness literature presents numerous mathematical notions and metrics, and also points to a tradeoff between them while satisficing some/all of them simultaneously. Furthermore, the contextual nature of fairness notions makes it difficult to automate bias evaluation in diverse algorithmic systems. Therefore, in this paper, we propose a novel model called latent assessment model (LAM) to characterize binary feedback provided by human auditors, by assuming that the auditor compares the classifier's output to his/her own intrinsic judgment for each input. We prove that individual and/or group fairness notions are guaranteed as long as the auditor's intrinsic judgments inherently satisfy the fairness notion at hand, and are relatively similar to the classifier's evaluations. We also demonstrate this relationship between LAM and traditional fairness notions on three well-known datasets, namely COMPAS, German credit and Adult Census Income datasets. Furthermore, we also derive the minimum number of feedback samples needed to obtain probably approximately correct (PAC) learning guarantees to estimate LAM for black-box classifiers. Moreover, we propose a novel multi-attribute reputation measure to evaluate auditor's preference towards various fairness notions as well as sensitive groups. These guarantees are also validated using standard machine learning algorithms, which are trained on real binary feedback elicited from 400 human auditors regarding COMPAS.

## 1 Introduction

Several machine learning (ML) based systems have recently been reported as being discriminatory with respect to the sensitive attributes (e.g. race and gender) in a variety of application domains, such as recommender systems in criminal justice (Angwin et al., 2016), e-commerce services in online markets (Fisman & Luca, 2016), and life insurance premiums (Waxman, 2018). Most of these applications have outcomes that depend on subjective human judgements which vary drastically across different people. Typically, these outcome labels are obtained on a case-by-case basis from different types of individuals (typically subject matter experts who have rich experience in their respective fields.). However, experts also exhibit biases especially when it comes to subjective judgements. For instance, in criminal justice applications, one may argue that a 'just' outcome for a parole application by a Black applicant is to deny the bail since he/she has a high chance to recidivate back to crime since the applicant lives in an unsafe neighborhood. On the other hand, the same applicant could be presented with a bail if he/she has no prior counts and has a clean record in the past. This idea of justice where everyone should be treated fairly based on their own features/merit, and independent of how others are treated, is called *non-comparative fairness* (Feinberg, 1974). In a practical data collection process, outcome labels are elicited from a non-comparative fairness standpoint since each person is presented with one case at a time (for example, readers may refer to (Goel & Faltings, 2019)). However, these judgements are usually subjective and even experts may not always arrive at a consensus on one correct verdict. Since the labelers suffer from inherent biases, their labels are often subject to inquisition (Cooper et al., 2021). This dearth of correct labels invalidates the quantification of ML performance using *accuracy*, which measures the alignment of classifier's outcome with the correct label found in training data.

On the contrary, algorithmic fairness literature do not question the fidelity of training data and discusses the quantification of biases and tradeoffs with accuracy using *comparative fairness* notions (Mehrabi et al., 2021).

Comparative fairness notions primarily focus on issues such as equality, similarity and proportionality, which questions whether an individual is treated/serviced 'justly' in comparison to that received by others. Biases measured by comparative notions can be broadly classified into two types, namely *disparate treatment*, where different groups of people are treated differently; and *disparate impact*, where different groups of people obtain different proportions of outcomes. Individual fairness (Dwork et al., 2012) quantifies disparate treatment via comparing the similarity between two individuals based on a task-specific distance metric. On the other hand, group fairness notions (Caton & Haas, 2020; Chouldechova & Roth, 2018; Hardt et al., 2016b) quantify disparate impact by comparing the differences in outcome proportions (e.g. true positive, false positive rates, predictive positive value) across different sensitive groups. In addition to measuring comparative fairness in ML systems, various mitigation techniques have been proposed to alleviate these biases in the literature. These mechanisms are typically categorized into three types: (i) *pre-processing* techniques that modifies the input to the ML model (Kamiran & Calders, 2012), (ii) *in-processing* techniques that retrain the ML model (Kamishima et al., 2012; Zafar et al., 2017), and (iii) *post-processing* techniques that modify the output labels (Hardt et al., 2016b; Corbett-Davies et al., 2017). Each of these approaches either maximize accuracy (or some loss function) under prescribed comparative fairness constraints, or maximize comparative fairness under tolerable constraints on accuracy loss. Finally, a fundamental tradeoff between comparative fairness and accuracy has also been investigated based on which, a compromise in accuracy was proposed to obtain fairness improvements in the system. However, comparative fairness notions cannot be completely automated and demands human feedback to identify appropriate fairness notions for system evaluation. Heterogeneous stakeholders compete against each other to enforce their preferred fairness notions, all of which cannot be satiated simultaneously due to a fundamental tradeoff that exists between comparative fairness notions (Chouldechova, 2017; Kleinberg et al., 2016). This dogfight between various stakeholders regarding the selection of an appropriate comparative fairness metric is also context-dependent (Binns, 2018), due to differences in protected groups across applications. Therefore, several researchers have attempted to explore human perception of fairness to identify the appropriate comparative fairness notion. For more details, the reader may refer to Section 1.2.

The inability of comparative notions to enable a reliable algorithmic fairness analysis, combined with the desertion of non-comparative fairness perspective in the past, leads to a major predicament in the design of fair ML algorithms. To the best of our knowledge, this paper makes the first attempt to address this gap via modeling human auditors from a non-comparative fairness perspective, and develop a scalable method to evaluate high-dimensional ML algorithms. Furthermore, this paper also assumes that the auditor provides a binary feedback regarding system fairness since the auditor's intrinsic evaluation of labels is a subconscious phenomenon and is practically difficult to quantify in most applications.

## 1.1 Technical Contributions

The main contributions of the paper are four-fold. Firstly, a novel *latent assessment model* (LAM) was proposed to represent binary human feedback regarding the fairness of the system, from a noncomparative perspective. Unlike most of the past literature on human perception of fairness, the proposed model allows human auditors to provide their feedback at will, without enforcing any specific fairness notion artificially (Gillen et al., 2018). Secondly, we emphasize that the availability of non-comparative fairness does not necessarily dispense with the need of comparative fairness. We prove that a system/entity satisfies individual and/or group fairness notions if the auditor exhibits LAM in his/her fairness evaluations. We also show that converse holds true in the case of individual fairness. Since both the system and auditor rules are hidden, we compute PAC learning guarantees on algorithmic auditing based on binary feedback obtained from human auditors. Third, we introduce a novel multi-attribute reputation measure to evaluate auditor's inherent biases based on various comparative fairness notions as well as sensitive attributes. Lastly, we validate the relationships with comparative fairness notions on three real datasets, namely COMPAS, Adult Income Census and German credit datasets. Using the feedback data of 400 crowd workers collected by (Dressel & Farid, 2018), we compare various learning frameworks such as logistic regression, support vector machines (SVM) and decision trees to estimate auditor's intrinsic judgements and their feedback. We also measure the reputation of the crowd workers to evaluate auditor's preference towards various comparative fairness notions.

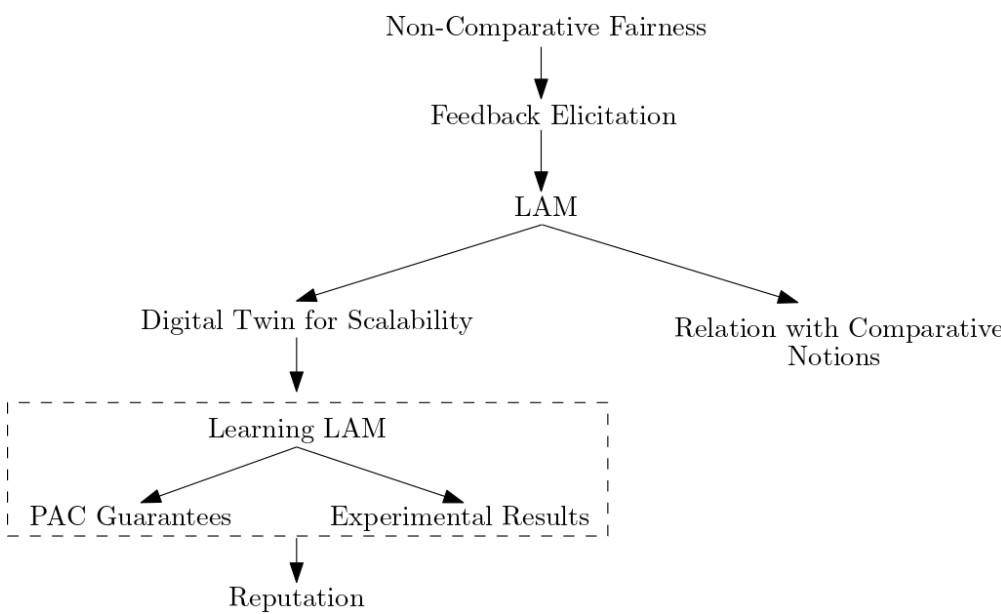

Figure 1: A Map of Technical Contributions

## 1.2 Related Work on Human Perception of Fairness and Worker Reputation

In the past, several researchers have attempted to model human perception of fairness, but have always tried to fit their revealed feedback to one of the existing fairness notions. For instance, task-based similarity metric used in individual fairness notion (Dwork et al., 2012) were estimated by (Jung et al., 2019) based on feedback elicited from auditors regarding how a given pair of individuals have been treated. Similarly, the work of (Gillen et al., 2018) assumes the existence of an auditor who is capable of identifying unfairness given pair of individuals when the underlying similarity metric is Mahalanobis distance. Saxena investigated how people perceive the fitness of three different individual fairness notions in the context of loan decisions (Saxena et al., 2019). The three notions are: (1) treat similar individuals similarly (Dwork et al., 2012), (2) Never favor a worse individual over a better one (Joseph et al., 2016), and (3) the probability of approval should be equal to the reward being the greatest (Liu et al., 2017). They show that people exhibit a preference for the last fairness definition.

From the perspective of group fairness notions, an experiment performed by (Srivastava et al., 2019) asks participants to choose among two different models to identify which notion of fairness (demographic parity or equalized odds) best captures people's perception in the context of both risk assessment and medical applications. Likewise, another team surveyed 502 workers on Amazon's Mturk platform and observed a preference towards *equal opportunity* in (Harrison et al., 2020). Note that both papers asked participants to reveal their analysis concerning a specific fairness notion in the context of given sensitive attributes (e.g. race, gender) which was clearly pointed out as a limitation of their work. On the contrary, in this paper, we impose no such restrictions on the auditor in constructing their feedback with respect to satiating any specific fairness notion. Instead, we assume that the expert auditor employs an intrinsic fair decision rule (which is unknown) to evaluate a given data tuple. Dressel and Farid in (Dressel & Farid, 2018) showed that COMPAS is as accurate and fair as that of untrained human auditors regarding predicting recidivism scores. On the other hand, (Yaghini et al., 2021) proposed a novel fairness notion, equality of opportunity (EOP), which requires that the distribution of utility should be the same for individuals with similar desert. Based on eliciting human judgments, they learned the proposed EOP notion in terms of criminal risk assessment context. Results show that EOP performs better than existing notions of algorithmic fairness in terms of equalizing utility distribution across groups. Another interesting work is by (Grgic-Hlaca et al., 2018), who discovered that people's fairness concerns are typically multi-dimensional (relevance, reliability, and volitionality), especially when binary feedback was elicited. This means that modeling human feedback should consider several factors beyond social discrimination. A major drawback of these approaches is that

the demographics of the participants involved in the experiments (Yaghini et al., 2021; Grgic-Hlaca et al., 2018; Harrison et al., 2020; Saxena et al., 2019) are not evenly distributed. For instance, the conducted experiments ask how models treated Caucasians and African-Americans. However, non-Caucasian participants were insufficient to assess the relationship between the participant's own demographics and the group that was disadvantaged by the model. Moreover, the participants are presented with multiple questions in the existing literature which cannot be scaled for larger decision-based models (Yaghini et al., 2021).

The problem of evaluating the biases of auditors/workers is also studied in crowdsourcing literature. Typically, auditors' reliability is also measured by comparing their responses with majority vote (Jamaludeen et al., 2019) or ground truth (Le et al., 2010). Since majority vote assumes that every auditor has same expertise, it cannot be applied to our framework as the biases of the auditors vary from one another. Moreover, due to the absence of ground truth, it is not possible to compare auditor responses. Recently, the work of Ghai et al. (2020) introduced a novel method of measuring worker's biases based on counterfactual fairness - a worker is considered to be fair if he/she provides the same label for an individual and its counterfactual. On the other hand, this paper proposed a novel reputation mechanism accounting various comparative fairness notions across different sensitive attributes.

## 2    Preliminaries: Traditional Fairness Notions

In most practical systems, two types of discrimination exist: *(i) disparate treatment*, where an individual is intentionally treated differently based on his/her membership in a protected class; and *(ii) disparate impact*, where members of a protected class are more negatively impacted than others. However, algorithmic fairness literature has studied a different set of fairness notions (ref. (Caton & Haas, 2020; Chouldechova & Roth, 2018; Mehrabi et al., 2021; Pessach & Shmueli, 2020)). Let $g(\cdot)$ be a ML system which predicts an outcome $y = g(x)$, where $x$ is the (multi-attribute) input variable and $y^*$ be the true label.

### 2.1    Group Fairness Notions

The notion of group fairness seek for parity of some statistical measure across all the protected attributes present in the data. Different versions of group-conditional metrics led to different group definitions of fairness. Let $A$ be the set of protected attributes, where $a \in A$ is the privileged group and $a' \in A$ is the underprivileged group.

**Statistical Parity:** This measures seeks to compute the probability difference of individuals who are predicted to be positive across different sensitive groups. Formally, it can be defined as followed.

$$\mathbb{P}[y = 1 \mid A = a] - \mathbb{P}[y = 1 \mid A = a'] \leq \delta \tag{1}$$

Since statistical parity aims for equal proportions of positive outcomes, the ideal value of probability difference should be 0. Specifically, if the difference is greater than 0, privileged group is benefited. Whereas, if the difference is less than 0, the underprivileged group is benefited. A major disadvantage of statistical parity is when the base rates (ratio of actual positive outcomes) are significantly different for various groups.

**Equal Opportunity:** To overcome the drawbacks in statistical parity, (Hardt et al., 2016a) introduced the notion of equalized odds which computes the difference between the true positive rates (TPR) of two protected groups.

$$\mathbb{P}[y = 1 \mid y^* = 1, A = a] - \mathbb{P}[y = 1 \mid y^* = 1, A = a'] \leq \delta. \tag{2}$$

Smaller differences between groups indicate better fairness. Since this notion considers the true label $y^*$, it assumes that the base rates of the two groups are representative and were not obtained in a biased manner.

**Calibration:** The measures computed the difference between positive predictive value of two groups. Positive predictive value represents the probability of an individual with a positive prediction actually experiencing a positive outcome. This notion is mathematically formulated as follows.

$$\mathbb{P}[y^* = 1 \mid y = 1, A = a] - \mathbb{P}[y^* = 1 \mid y = 1, A = a'] \leq \delta. \tag{3}$$

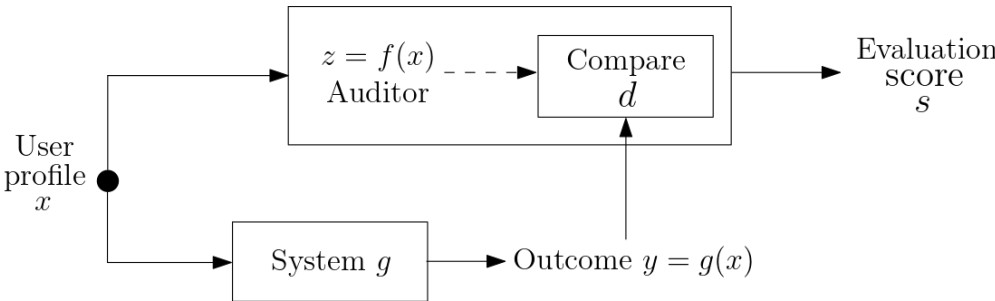

Figure 2: Latent Assessment Model of the Expert Auditor

Although in some cases equal calibration may be the desired measure, it has been shown that it is incompatible with equalized odds (Pleiss et al., 2017) and is insufficient to ensure accuracy (Corbett-Davies & Goel, 2018).

**Equal Accuracy:** This requires similar accuracy across groups (Berk et al., 2018).

$$\mathbb{P}[y = y^* \mid A = a] - \mathbb{P}[y = y^* \mid A = a'] \leq \delta. \tag{4}$$

## 2.2 Individual Fairness

Individual fairness assessments, rather than measuring discrimination across different sensitive groups, consider fairness for each individual, with the assumption that similar individuals should be treated as similarly as feasible (Dwork et al., 2012). Formally, given any two individuals $x_i, x_j \in \mathcal{X}$ and for some $\kappa, \delta \geq 0$, the ML system $g$ is $(\kappa, \delta)$-individually fair if

$$d(y_i, y_j) \leq \delta, \quad \text{if } \mathcal{D}(x_i, x_j) \leq \kappa, \tag{5}$$

where $y_i = g(x_i)$ and $y_j = g(x_j)$ are the system's outputs for the inputs $x_i$ and $x_j$ respectively.

Note that the above definition contains a slack in the bound presented in Equation 5. Since the original definition of individual fairness directly compares the two similarlity metrics $d(y_i, y_j)$ and $\mathcal{D}(x_i, x_j)$, it becomes very challenging to find appropriate distance metrics in practice. Furthermore, this relaxed definition enables us to evaluate a relationship between our fairness model and individual fairness. Such relaxations have also been considered in the past for rigorous analysis John et al. (2020).

Unfortunately, in most practical applications, the similarity metric $\mathcal{D}(x_i, x_j)$ is task-specific and is typically unknown, which causes a severe restraint on our ability to ensure individual fairness. As a solution, metric learning was proposed to discern a task-specific similarity metric by evaluating the relative distance between human judgements (Ilvento, 2019) for any given pair of inputs. On the other hand, (Mukherjee et al., 2020) utilizes Mahalanobis distance as a fair metric and proposed EXPLORE, an algorithm to learn similarity between individuals from pairs of comparable and incomparable samples. It learns similarity such that the logistic regression predicts "comparable" when the fair distance is short, and "incomparable" when the fair distance is large. Interested readers can refer to (Fleisher, 2021) which discussed various inefficiencies of individual fairness in detail.

## 3  Latent Assessment Model and Non-Comparative Evaluation of Human Auditors

Consider an expert auditor who is presented with a data tuple $(x, y)$, where $x \in \mathcal{X}$ is the input given to ML model $g$ and $y = g(x) \in \mathcal{Y}$ is the output label as shown in Figure 2. Let $f$ denote the expert auditor's latent decision rule, $z = f(x)$ be the latent subjective label for the input $x$, and $s(x, y, z)$ represent the auditor's binary feedback regarding the input-output pair $(x, y)$. In this paper, we model auditor's judgments from a non-comparative perspective as follows:

**Definition 1** ($\epsilon$-Latent Assessment Model)**.** *An auditor is said to satisfy $\epsilon$-LAM if there exists a tuple* $(\mathcal{X}, \mathcal{Y}, d, f, \epsilon)$ *such that the auditor compares the system's output $y = g(x)$ with the latent subjective label*

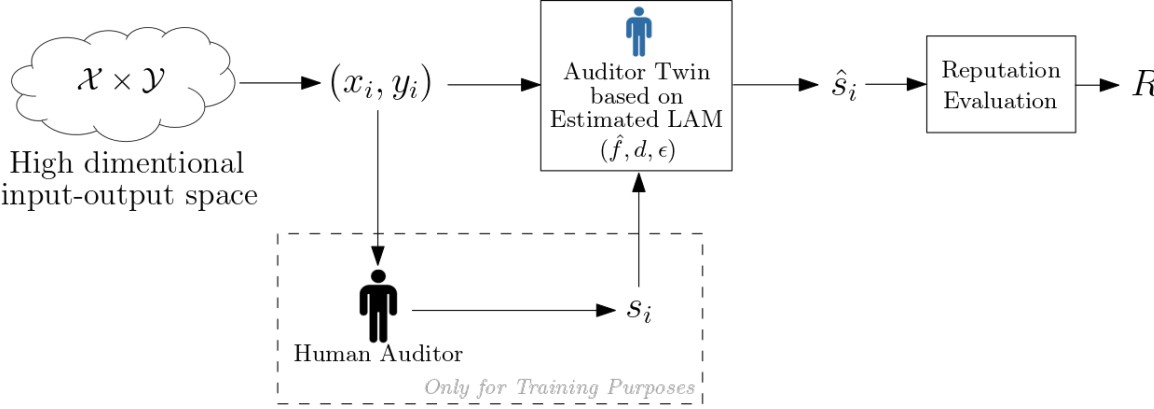

Figure 3: Auditor Evaluation Framework based on $\epsilon$-LAM

*$z = f(x)$ using a distance metric d and reveal his/her binary feedback as*

$$s(x, y, z) = \begin{cases} 1, & if \ \ d(y, z) \geq \epsilon, \\ 0, & otherwise. \end{cases} \tag{6}$$

*Furthermore, we say the auditor f is $\epsilon$-LAM with respect to the system g if $d(y, z) < \epsilon$ for all $x \in \mathcal{X}$.*

*We also say that the auditor f is $\epsilon$-LAM with respect to the system g with high probability, if $\mathbb{P}\Big(d(y, z) < \epsilon\Big) \geq 1 - \delta$, for some small $\epsilon > 0$ and $\delta > 0$.*

In other words, if the distance between the output label $y = g(x)$ and the latent subjective label $z = f(x)$ is greater than $\epsilon$, the auditor will deem the system label $y = g(x)$ as *unfair.* For the sake of illustration, consider an individual who committed felony and has multiple prior offences, received low recidivism score from a risk assessment tool. The expert auditor evaluates the individual intrinsically and may decide that he/she should receive a higher recidivism score. Then, the auditor judges the tool's output as unfair. In this paper, we assume the fair relation $f$ employed by the expert auditor is unknown. Therefore, we need to learn the proposed $\epsilon$-LAM using statistical learning techniques. This will be discussed in Section 4 in detail. Although $\epsilon$-LAM comprises of three unknowns in practice, namely $d$, $f$ and $\epsilon$, we assume that the distance metric $d$ used by the auditor is known.

**Remark 1.** *In the case of binary classification, the Hamming distance between g and f, denoted as $d_H\big(y = g(x), z = f(x)\big)$, takes a binary value of 0 or 1. Therefore, for any $0 < \epsilon < 1$, $\epsilon$-LAM model in Definition 1 reduces to $s = y \oplus z$, where $\oplus$ denotes the XOR operation. Since the XOR function is reversible, we can infer the latent subjective label $z = y \oplus s$ exactly from elicited feedback.*

Note that auditors exhibit different types of biases in the real-world. Examples include confirmation bias, hindsight bias, anchoring bias, racial and gender bias. Every auditor is susceptible to biases which are shaped by their prior experiences and/or knowledge about various social groups within the community. Consequently, it is not reasonable to accept any auditor's feedback blindly without evaluating their inherent biases. Our proposed model, LAM, captures auditor's biases through the latent subjective label $z = f(x)$ in Equation 6. However, in order to quantify auditor biases, we need to investigate the relationship between LAM and comparative fairness notions. By doing so, we can estimate any given auditor's performance in terms of diverse fairness notions that do not necessarily align with each other.

In this paper, we propose a novel auditor evaluation framework for high-dimensional ML systems in Figure 3 to quantify his/her biases across diverse comparative fairness notions. Since it is impractical to elicit feedback that spans entirely across the high-dimensional input space, there is a need to estimate LAM for a given auditor to predict his/her fairness evaluation for any input-output pair $(x, y) \in \mathcal{X} \times \mathcal{Y}$. In this paper, we assume that the distance function $d(\cdot)$ and $\epsilon$ are known and restrict our attention to learning

the auditor's latent decision rule $f$ from limited elicited feedback. In practice, human auditors are typically incapable of expressing their intrinsic evaluations $f$ formally in functional form. Therefore, we predict auditor's latent subjective labels $\hat{z} = \hat{f}(x)$ on any input $x \in \mathcal{X}$ using a learning algorithm $\mathcal{A}$ (e.g. logistic regression, random forests, and support vector machines), which is trained using elicited feedback samples $\{(x_1, y_1, s_1), \cdots, (x_n, y_n, s_n)\}$. Furthermore, most ML-based systems are not revealed to fairness evaluators, and need to be assessed as black-box models. For example, the model used to develop COMPAS is proprietary and is not revealed to any criminal justice evaluator. In other words, it is also essential to estimate the system model $\hat{g}$ from available data $\{(x_1, y_1), \cdots, (x_n, y_n)\}$, and predict its output label $\hat{y} = \hat{g}(x)$.

Using the two predictions $\hat{y}$ and $\hat{z}$, we can now predict the binary feedback

$$\hat{s}(x, \hat{y}, \hat{z}) = \begin{cases} 1, & \text{if } d(\hat{y}, \hat{z}) \geq \epsilon, \\ 0, & \text{otherwise.} \end{cases} \tag{7}$$

Then, we measure the overall bias in the system from a noncomparative fairness perspective as defined below.

**Definition 2** (Fairness Measure). *The direct empirical fairness measure (DEFM) from a noncomparative fairness perspective is defined formally as the average feedback obtained directly from the original auditor, i.e.*

$$\mu = \mathbb{E}(s) = \frac{1}{n} \sum_{i=1}^{n} s_i. \tag{8}$$

*On the other hand, the indirect empirical fairness measure (IEFM) from a noncomparative fairness perspective is given by the empirical estimate of the true bias, i.e.*

$$\hat{\mu} = \mathbb{E}(\hat{s}) = \frac{1}{n} \sum_{i=1}^{n} \hat{s}_i, \tag{9}$$

*where $\hat{s}$ is computed according to Equation 7.*

The following section evaluates theoretical guarantees in terms of sample complexity and fundamental limits of the proposed auditor evaluation framework due to the two predictors $\hat{f}$ and $\hat{g}$.

## 4 PAC Guarantees

Most ML systems map high-dimensional input spaces to low-dimensional outcomes. For example, in criminal justice or banking applications, input space spans across the entire population of individuals serviced by the ML system, while the outcome space is finite (e.g. binary decisions such as approval/disapproval of bails or loans.). In such a scenario, feedback elicitation from human auditors is extremely challenging due to the large number of feasible input-output pairs, which cannot be evaluated by any one human auditor. Therefore, there is a need for human-system teaming for a practically scalable fairness evaluation, wherein human auditors give feedback for a small subset of input-output pairs and the system learns their LAM from elicited feedback and scales it across all other possibilities for a wholesome evaluation. This section investigates the fundamental limits of such fairness evaluation systems in terms of sample complexity (i.e. the minimum number of feedback samples needed to effectively learn auditor's LAM with prescribed guarantees) as well as the evaluation performance.

Let $\mathcal{C}$ be a class of auditor's decision rules $f : \mathcal{X} \rightarrow \mathcal{Y}$ where $\mathcal{X}$ is the set of all possible individuals serviced by the ML system and $\mathcal{Y}$ is the set of output labels of the ML system. Let $\mathcal{A}$ denote a learning algorithm that identifies a hypothesis $\hat{f}_n \in \mathcal{H}$ that minimizes the loss $d(f, \hat{f}_n)$, based on $n$ observations $\{(x_i, y_i, s_i)\}_{i=1}^{n} \in \mathcal{X}^n \times \mathcal{Y}^n \times \{0, 1\}^n$, which follow some data distribution $D$. Note that the hypothesis class $\mathcal{H}$ is not necessarily equal to the original class of decision rules $\mathcal{C}$ due to our inability to mathematically characterize $\mathcal{C}$ exactly, especially due to the complexities in human decision making. The PAC model provides guarantees for minimum number of such instances needed to obtain a hypothesis $\hat{f}_n$ with error no more than $\epsilon$ with probability $1 - \delta$. Formally,

**Definition 3** (PAC Learnability of $f$). *We say that the auditor's latent decision rule $f$ is PAC-learnable (in both realizable and non-realizable settings) if there exists numbers $\epsilon_f > 0$, $0 < \delta_f < 1$, $N_f(\epsilon_f, \delta_f) > 0$, and an algorithm $\mathcal{A}_f$ which receives $n \geq N_f(\epsilon_f, \delta_f)$ i.i.d. samples (drawn from some distribution $D$) as input, and outputs an estimated classifier $\hat{f}_n$ such that*

$$\mathbb{P}\Big(d(f, \hat{f}_n) \leq \epsilon_f\Big) \geq 1 - \delta_f.$$

Similarly, the ML-based system is unavailable to the evaluation platform i.e. $g$ in unknown.

**Definition 4** (PAC Learnability of $g$). *We say that the ML-based system $g$ is PAC-learnable (in both realizable and non-realizable settings) if there exists numbers $\epsilon_g > 0$, $0 < \delta_g < 1$, $N_g(\epsilon_g, \delta_g) > 0$, and an algorithm $\mathcal{A}_g$ which receives $n \geq N_g(\epsilon_g, \delta_g)$ i.i.d. samples (drawn from some distribution $D$) as input, and outputs an estimated classifier $\hat{g}_n$ such that*

$$\mathbb{P}\Big(d(g, \hat{g}_n) \leq \epsilon_g\Big) \geq 1 - \delta_g.$$

As discussed earlier, the auditor's intrinsic rule $f$ is not directly revealed, while eliciting his/her feedback regarding biases in the system-of-interest. Therefore, we need to compute the intrinsic rule $\hat{f}_n$ in order to reproduce auditor's judgements for other input possibilities. At the same time, the classifier is typically unavailable to the bias-evaluation platform, i.e., $g$ is also unknown to the bias-evaluation platform. In other words, a practical bias-evaluation platform is expected to learn $(f, g)$ to obtain a model $(\hat{f}_n, \hat{g}_n)$, and identify an appropriate fairness notion for the given context so that the bias evaluation platform can algorithmically evaluate bias in a system with a large input space. In the following theorem, we provide guarantees for the algorithmic $\epsilon - LAM$ evaluations, based on estimated rules $\hat{f}_n$ and $\hat{g}_n$.

**Theorem 1.** *Let $N$ denote the minimum number of samples needed to guarantee $\epsilon - LAM$ empirically, i.e. $\mathbb{P}\Big(d(\hat{g}_n, \hat{f}_n) < \epsilon\Big) > 1 - \delta$ for some $\epsilon > 0$ and $\delta > 0$. Then, for any auditor's intrinsic rule $f$ and classifier $g$, there exists some $0 < N_f, N_g < N$, $\epsilon_g, \epsilon_f, \epsilon, \tilde{\delta}, \delta_g, \delta_f > 0$ such that $\epsilon_g + \epsilon_f + \epsilon < \tilde{\epsilon}$ and $\delta_g + \delta_f + \delta < 2 + \tilde{\delta}$, and an algorithm $\mathcal{A}$ that receives i.i.d. samples $\{(x_i, y_i, z_i)\}_{i=1}^n$ as input, and outputs rules $\hat{f}_n$ and $\hat{g}_n$ with a probability of $d(f(x), g(x)) < \epsilon$ being at least $1 - \tilde{\delta}$, only when*

$$n \geq N \triangleq \min_{\epsilon_1, \epsilon_2, \delta_1, \delta_2} (\max\{N_g, N_f\}), \tag{10}$$

*where $N_f$ and $N_g$ satisfy PAC learning bounds for $f$ and $g$.*

*Proof.* In this proof, we will show that $f$ is $\epsilon$-LAM with respect to $g$ with high probability, i.e.

$$\mathbb{P}\Big(d(f(x), g(x)) < \epsilon\Big) \geq 1 - \delta \tag{11}$$

for some small $\epsilon > 0$ and $\delta > 0$. Assuming that both $f$ and $g$ are PAC-learnable functions (as stated in Definitions 3 and 4 respectively), we evaluate the triangle inequality between the labels $z = f(x)$, $y = g(x)$, $\hat{z} = \hat{f}_n(x)$ and $\hat{y} = \hat{g}_n(x)$, as shown below:

$$
\begin{aligned}
d(g(x), f(x)) &\leq d(g(x), \hat{g}_n(x)) + d(\hat{g}_n(x), f(x)) &&\text{(due to } \Delta \text{ ineq. between } y, \hat{y} \text{ and } z) \\[6pt]
&\leq d(g(x), \hat{g}_n(x)) + d(\hat{g}_n(x), \hat{f}_n(x)) + d(\hat{f}_n(x), f(x)) &&\text{(due to } \Delta \text{ ineq. between } \hat{y}, \hat{z} \text{ and } z).
\end{aligned}
\tag{12}
$$

Note that $f$ and $g$ are PAC-learnable functions with parameters $(\epsilon_f, \delta_f, N_f)$ and $(\epsilon_g, \delta_g, N_g)$ respectively. Let $\hat{f}$ be $\hat{\epsilon}$-LAM with respect to $\hat{g}$ with high probability, i.e.

$$\mathbb{P}\Big(d(\hat{f}(x), \hat{g}(x)) < \hat{\epsilon}\Big) \geq 1 - \hat{\delta}, \tag{13}$$

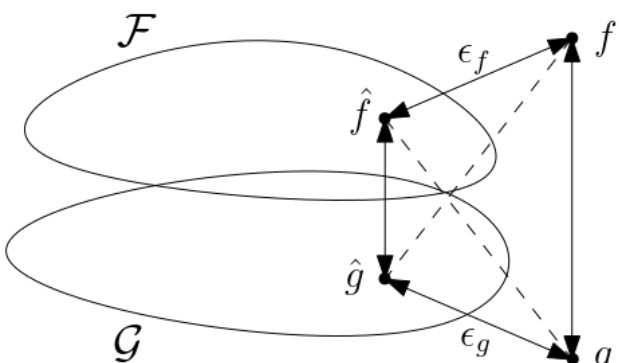

Figure 4: The interaction between the learning algorithms of $f$ and $g$.

for some $\hat{\epsilon} > 0$ and $\hat{\delta} > 0$. Then, we can always find some number $\epsilon > 0$ such that $\epsilon_g + \hat{\epsilon} + \epsilon_f \leq \epsilon$. In other words, if we define three events: $\mathcal{E}_g \doteq \{d(g(x), \hat{g}_n(x)) < \epsilon_g\}$, $\mathcal{E}_{LAM} \doteq \{d(\hat{g}_n(x), \hat{f}_n(x)) < \hat{\epsilon}\}$ and $\mathcal{E}_f \doteq \{d(f(x), \hat{f}_n(x)) < \epsilon_f\}$, the intersection of these events will guarantee

$$
\begin{aligned}
d(g(x), f(x)) &\leq d(g(x), \hat{g}_n(x)) + d(\hat{g}_n(x), \hat{f}_n(x)) + d(\hat{f}_n(x), f(x)) \\
&\leq \epsilon_g + \hat{\epsilon} + \epsilon_f \qquad\qquad\qquad (14)\\
&< \epsilon.
\end{aligned}
$$

Therefore, from Frechet's lower bound on the probability of intersection of events, we obtain

$$
\begin{aligned}
\mathbb{P}\Big(d(g(x), f(x)) < \epsilon\Big) &\geq \mathbb{P}\Big(\mathcal{E}_g \wedge \mathcal{E}_{LAM} \wedge \mathcal{E}_f\Big) \\
&\geq \mathbb{P}(\mathcal{E}_g) + \mathbb{P}(\mathcal{E}_{LAM}) + \mathbb{P}(\mathcal{E}_f) - 2, \qquad \text{(due to Frechet's lower bound)} \\
&\geq (1 - \delta_g) + (1 - \hat{\delta}) + (1 - \delta_f) - 2 \qquad\qquad\qquad (15)\\
&> 1 - \delta,
\end{aligned}
$$

where $\delta$ is some number such that $\delta_g + \hat{\delta} + \delta_f < \delta$.

Let $N_g(\epsilon_g, \delta_g)$ and $N_f(\epsilon_f, \delta_f)$ denote the minimum samples needed to guarantee PAC learnability at $g$ and $f$ respectively. In other words, the maximum of the two numbers will guarantee PAC learnability of both $g$ and $f$, i.e. $N(\epsilon_g, \epsilon_f, \delta_g, \delta_f) = \max\{N_g(\epsilon_g, \delta_g), N_f(\epsilon_f, \delta_f)\}$. However, there are several possible ways to split the prescribed tolerance $\epsilon$ and $\delta$ across the terms $(\epsilon_f, \epsilon_g, \hat{\epsilon})$ and $(\delta_f, \delta_g, \hat{\delta})$ such that the two inequalities $\epsilon_g + \hat{\epsilon} + \epsilon_f < \epsilon$ and $\delta_g + \hat{\delta} + \delta_f < \delta$ hold true. In other words, the minimum number of samples needed to achieve prescribed guarantees $(\epsilon, \delta)$ as stated in Equation 11 is given by Equation 10. □

Note that IEFM $\hat{\mu}$ cannot be better than DEFM $\mu$ due to the estimation errors in $\hat{f}_n$ and $\hat{g}_n$. In other words, there is no free lunch in terms of fairness measurement when auditor and/or system are replaced by digital twins. However, if the learning errors in both $\hat{f}_n$ and $\hat{g}_n$ are small, we demonstrate that there is a tolerable gap between the direct and indirect fairness measurement in Section 8 (ref. to Figure 6). Since the problem of finding the gap between DEFM and IEFM remained intractable, we present an upper bound on the distance $d(\hat{f}_n(x), \hat{g}_n(x))$ in terms of $d(f(x), g(x))$, as a surrogate analysis.

**Theorem 2.** *If $\mathcal{A}$ is a learning algorithm which receives a training set of size $n \geq N \triangleq \min_{\epsilon_1, \epsilon_2, \delta_1, \delta_2} (\max\{N_g, N_f\})$, there always exists a distribution $D$ over $\mathcal{X}$ such that*

$$d(\hat{f}_n(x), \hat{g}_n(x)) \le \epsilon_f + \epsilon_g + d(f(x), g(x)),$$

*for any $x \in \mathcal{X}$.*

*Proof.* We illustrate this result based on the following scenario. Consider that the true LAM and the estimated LAM have opposing opinion regarding an input $x$, i.e., $s(x, y, \hat{y}) = 1$ and $\hat{s}(x, z, \hat{z}) = 0$. In other words, we assume that $d(f(x), g(x)) \ge \epsilon$ and $d(\hat{f}_n(x), \hat{g}_n(x)) < \epsilon$. The objective is to measure the deviation between the auditor's true LAM and the estimated auditor twin's LAM. Figure 4 depicts this scenario, where $\mathcal{F}$ and $\mathcal{G}$ denote the realizable function spaces from which we choose the estimates $\hat{f}_n \in \mathcal{F}$ and $\hat{g}_n \in \mathcal{G}$ of the auditor's intrinsic rule $f$ and the black-box classifier $g$ respectively. Since $f$ and $g$ are unknown, it is reasonable to assume that the two functions do not belong to the function spaces $\mathcal{F}$ and $\mathcal{G}$ respectively. Let $\epsilon_f$ and $\epsilon_g$ denote the errors in estimating the functions $\hat{f}_n$ and $\hat{g}_n$ respectively.

Based on triangle inequality, the following can be inferred.

$$d(\hat{f}_n(x), \hat{g}_n(x)) \le d(f(x), \hat{f}_n(x)) + d(f(x), \hat{g}_n(x)) \le \epsilon_f + d(f(x), \hat{g}_n(x)), \tag{16}$$

since the learning algorithm used to estimate $f$ comes with a bias of $\epsilon_f$, i.e. we have $d(f, \hat{f}_n) \le \epsilon_f$. Similarly, we have

$$d(\hat{f}_n(x), \hat{g}_n(x)) \le d(g(x), \hat{g}_n(x)) + d(\hat{f}_n(x), g(x)) \le \epsilon_g + d(\hat{f}_n(x), g(x)). \tag{17}$$

Adding above two equations we obtain

$$d(\hat{f}_n(x), \hat{g}_n(x)) \le \frac{1}{2}\left(\epsilon_f + \epsilon_g + d(\hat{f}_n(x), g(x)) + d(f(x), \hat{g}_n(x))\right). \tag{18}$$

Furthermore, we have

$$
\begin{aligned}
d(f(x), \hat{g}_n(x)) &\le \epsilon_g + d(f(x), g(x)) \\
d(\hat{f}_n(x), g(x)) &\le \epsilon_f + d(f(x), g(x)).
\end{aligned}
\tag{19}
$$

Substituting Equation 19 in Equation 18, we obtain

$$d(\hat{f}_n(x), \hat{g}_n(x)) \le \frac{1}{2}\left(\epsilon_f + \epsilon_g + \epsilon_g + d(f(x), g(x)) + \epsilon_f + d(f(x), g(x))\right) \le \epsilon_f + \epsilon_g + d(f(x), g(x)). \tag{20}$$

$\square$

# 5 Relationship Between Non-Comparative and Comparative Fairness Notions

Although comparative and non-comparative fairness notions are fundamentally different from each other, this section discusses their mutual relationship formally.

## 5.1 Relation with Individual Fairness

In the following proposition, we show how the system $g$ can be evaluated based on the notion of $(\kappa, \delta)$-individual fairness, when the auditor $f$ is $\epsilon$-LAM with respect to the system $g$.

**Proposition 1.** *A system $g$ is $(\kappa, 2\epsilon + \delta)$-individually fair, if its auditor $f$ is $(\kappa, \delta)$-individually fair and satisfies $\epsilon$-LAM with respect to the system $g$.*

*Proof.* Given $(x_i, y_i)$ and $(x_j, y_j) \in \mathcal{X} \times \mathcal{Y}$ such that $\mathcal{D}(x_i, x_j) \le \kappa$ (the two individuals are $\kappa$-similar), then $f$ is $(\kappa, \delta)$-individually fair if $d\left(f(x_i), f(x_j)\right) < \delta$. However, note that if $g$ is $\epsilon$-LAM with respect to $f$, then

$d\Big(g(x_i), f(x_i)\Big) < \epsilon$ and $d\Big(g(x_j), f(x_j)\Big) < \epsilon$. Therefore, by applying a chain of triangle inequalities, we obtain

$$
\begin{aligned}
d\Big(g(x_i), g(x_j)\Big) &\leq d\Big(g(x_i), f(x_i)\Big) + d\Big(f(x_i), f(x_j)\Big) + d\Big(f(x_j), g(x_j)\Big) \\
&< 2\epsilon + \delta,
\end{aligned}
\tag{21}
$$

for all $x_i, x_j$ such that $\mathcal{D}(x_i, x_j) \leq \kappa$. $\qquad\square$

**Remark 2.** *In the case of binary classifiers, if the auditor $f$ is $\kappa$-individually fair, we have $f(x_i) = f(x_j)$ for all $x_i, x_j$ whenever $\mathcal{D}(x_i, x_j) \leq \kappa$. Furthermore, if the auditor $f$ is $\epsilon$-LAM with respect to the system $g$ for any $\epsilon \in (0, 1)$, we have $g(x) = f(x)$ for all $x \in \mathcal{X}$. Combining the above two properties, we get $g(x_i) = f(x_j) = f(x_i) = g(x_j)$ for all $x_i, x_j$ such that $\mathcal{D}(x_i, x_j) \leq \kappa$.*

We illustrate this result using the following example from the banking domain. Consider two individuals who are looking to apply for a loan. A banking system would evaluate both the applications via collecting information such as gender, race, address, credit history, collateral, and his/her ability to pay back. At the same time, consider an auditor who makes fairness judgements based on the rule: "If he/she has cleared all the debts and possesses reasonably valued collateral, the loan must be granted". Given that the auditor treats any two similar individuals similarly, the auditor satisfies individual fairness. Hence, if the evaluation of the banking system is relatively similar to the auditor's fair relation, from Proposition 1, the banking system is also individually fair.

**Proposition 2.** *If an auditor $f$ is not $(\kappa, \delta)$-individually fair, but satisfies $\epsilon$-LAM with respect to the system $g$, then the system is not $(\kappa, \delta - 2\epsilon)$-individually fair as well.*

*Proof.* If $f$ is not individually fair, then for some input pair $(x_i, x_j)$ such that $\mathcal{D}(x_i, x_j) < \kappa$, we have $d(f(x_i), f(x_j)) > \delta$ for all $\kappa, \delta \in \mathbb{R}$. However, note that if $g$ is $\epsilon$-LAM with respect to $f$, then $d(g(x_i), f(x_i)) < \epsilon$ and $d(g(x_j), f(x_j)) < \epsilon$. Therefore, by applying a chain of triangle inequalities, we have

$$
d(f(x_i), f(x_j)) \leq d(g(x_i), f(x_i)) + d(g(x_j), f(x_j)) + d(g(x_i), g(x_j))
\tag{22}
$$

Substituting the bounds of $d(g(x_j), f(x_j))$ and $d(g(x_i), g(x_j))$ we get

$$
\begin{aligned}
2\epsilon + d(g(x_i), g(x_j)) &> d(g(x_i), f(x_i)) + d(g(x_j), f(x_j)) + d(g(x_i), g(x_j)) \\
&\geq d(f(x_i), f(x_j)) \\
&> \delta
\end{aligned}
\tag{23}
$$

for all $\delta \in \mathbb{R}$. Therefore, we also have $d(g(x_i), g(x_j)) > \delta - 2\epsilon$. $\qquad\square$

**Remark 3.** *For binary classification, Proposition 2 can be reduced as follows. Note that if $f$ is not $\kappa$-individually fair, we have $f(x_i) \neq f(x_j)$ even though $\mathcal{D}(x_i, x_j) \leq \kappa$. Furthermore, if $g$ is $\epsilon - LAM$ with respect to $f$ for any $\epsilon \in (0, 1)$, we have $g(x) = f(x)$ for all $x \in \mathcal{X}$. Combining the above two properties, we get $g(x_i) = f(x_j) \neq f(x_i) = g(x_j)$ for all $x_i, x_j$ such that $\mathcal{D}(x_i, x_j) \leq \kappa$.*

Consider the earlier example of banking where, there are two individuals, $A$ and $B$, who possess the same degree of merit. Imagine that the bank approves $A$'s loan application and denies $B$. This outcome remains the same as per the auditor's fair relation. Imagine further that neither $A$ nor $B$ merits the outcome. Though both banking's evaluation and auditor's judgements seem to be similar, they violate the precept, "treat similar individuals similarly". Moreover, $A$ is treated in a way that $A$ does not merit. Hence, we can assert that banking evaluation does not satisfy individual fairness.

### 5.2 Relation with Group Fairness Notions

As discussed earlier, group fairness notions compare certain probabilistic measure across two protected groups. In the remaining section, we will focus on the relationship between group fairness notions and our proposed LAM. For the sake of convenience, let us denote $p_{x,y}(g,a) = \mathbb{P}[g(x) = y \mid A = a]$.

**Proposition 3.** *Given that the probability distributions are $M$-Lipschitz continuous over all possible $f$ and $g$ functions, $g$ satisfies $(2M\epsilon + \delta)$-statistical parity, if $g$ is $\epsilon$-LAM with respect to $f$, and $f$ satisfies $\delta$-statistical parity.*

*Proof.* Given the set of protected attributes $\mathcal{A}$, since $f$ satisfies $\delta$-statistical parity, we have $||p_{x,y}(f,a) - p_{x,y}(f,a')|| < \delta$ for all $a, a' \in \mathcal{A}$. Then, we have

$$p_{x,y}(g,a) - p_{x,y}(g,a') = [p_{x,y}(g,a) - p_{x,y}(f,a)] + [p_{x,y}(g,a') - p_{x,y}(f,a')] \tag{24}$$
$$+ [p_{x,y}(f,a) - p_{x,y}(f,a')]$$

Assuming $M$-Lipschitz continuity over all $f(x)$, $g(x)$, we have $||p_{x,y}(g,a) - p_{x,y}(f,a)|| < M \cdot \epsilon$, since $d(g(x), f(x)) < \epsilon$. Combining all the inequalities, we have

$$||p_{x,y}(g,a) - p_{x,y}(g,a')|| < 2M\epsilon + \delta. \tag{25}$$

$\square$

Again, consider the earlier example of loan approvals to illustrate the above proposition. Consider that there exists two groups which are classified based income - low and high. The banking system builds a credit model based purely. Moreover, the system may decide to use different requirement levels - low interest or default to low income group, so that the percentage of people getting a loan in low-income group is equal to the percentage of people getting a loan in high-income group. Now, suppose an auditor presents fair judgements based on the rule: "If Group A has a FICO credit score of 550 and cleared all the debts, the loan must be granted. If Group B has a FICO score of 700 and has valuable collateral, grant the loan". Note that, the auditor's fair relation is somewhat similar to that of the bank's policy. Since the bank's policy is known to be statistically fair, the auditor is also unbiased from a group fairness perspective.

Similarly, the following three propositions identify the relationship between our proposed $\epsilon - LAM$ and three other group fairness notions, namely equal opportunity, calibration, and equal accuracy.

**Proposition 4.** *Given that the probability distributions are $M$-Lipschitz continuous over all possible $f$ and $g$ functions, $g$ satisfies $(2M\epsilon + \delta)$-equal opportunity, if $g$ is $\epsilon$-LAM with respect to $f$, and $f$ satisfies $\delta$-equal opportunity.*

*Proof.* The proof is similar to that of Proposition 3. Therefore, for the sake of brevity, the proof is not included. $\square$

**Proposition 5.** *Given that the probability distributions are $M$-Lipschitz continuous over all possible $f$ and $g$ functions, $g$ satisfies $(2M\epsilon + \delta)$-calibration, if $g$ is $\epsilon$-LAM with respect to $f$, and $f$ satisfies $\delta$-calibration.*

*Proof.* The proof is similar to that of Proposition 3. Therefore, for the sake of brevity, the proof is not included. $\square$

**Proposition 6.** *Given that the probability distributions are $M$-Lipschitz continuous over all possible $f$ and $g$ functions, $g$ satisfies $(2M\epsilon + \delta)$-equal accuracy, if $g$ is $\epsilon$-LAM with respect to $f$, and $f$ satisfies $\delta$-equal accuracy.*

*Proof.* The proof is similar to that of Proposition 3. Therefore, for the sake of brevity, the proof is not included. $\square$

# 6 Comprehensive Reputation of Human Auditors

Given the auditor's intrinsic labels $\{\hat{z}_1, \cdots, \hat{z}_K\}$, we can compute the auditor's performance for a given sensitive/protected group in terms of various fairness notions such as statistical parity (ref. Equation equation 1), equal opportunity (ref. Equation equation 2), calibration (ref. Equation equation 3) and individual fairness (ref. Section 2.2). Furthermore, note that this multi-dimensional fairness evaluation is different for different sensitive groups. For example, in the United States, protected groups are typically defined based on race, gender, religion or any combination of these attributes. Such multi-attribute fairness evaluations across different sensitive/protected groups naturally steers us towards defining a multi-attribute reputation matrix

$$R(\hat{z}_1, \cdots, \hat{z}_K) = \begin{bmatrix} r_{1,1}(\hat{z}_1, \cdots, \hat{z}_K) & \cdots & r_{1,L}(\hat{z}_1, \cdots, \hat{z}_K) \\ \vdots & \ddots & \vdots \\ r_{M,1}(\hat{z}_1, \cdots, \hat{z}_K) & \cdots & r_{M,L}(\hat{z}_1, \cdots, \hat{z}_K) \end{bmatrix}, \tag{26}$$

where $M$ is the total number of fairness notions and $L$ is the total number of sensitive groups. For example, if we are evaluating the auditor based on *statistical parity (SP)* with respect to the sensitive attribute *race*, then $r_{SP,\ race} = \mathbb{P}[f(x) = 1 \mid race = a] - \mathbb{P}[f(x) = 1 \mid race = a']$.

Although we propose a multi-dimensional fairness evaluation in a matrix format $R$, it is necessary to represent auditor's biases as a one-dimensional score $\nu$ signifying his/her overall performance with respect to various fairness notions. Therefore, we apply Frobenius norm of the reputation matrix $R$ to compute the auditor's scalar reputation score as follows.

$$\nu(\hat{z}_1, \cdots, \hat{z}_K) = ||R(\hat{z}_1, \cdots, \hat{z}_K)||_F = \sqrt{\sum_{i=1}^{M} \sum_{j=1}^{L} \left| r_{i,j}(\hat{z}_1, \cdots, \hat{z}_K) \right|^2}. \tag{27}$$

We choose this reputation score due to the following axiomatic properties:

- **Perfect Fairness:** A utopian auditor satisfies all the fairness notions, i.e. every entry in $R$ becomes zero. Consequently, $||R||_F \to 0$.

- **Lipschitz-Boundedness:** Consider any deviation $\Delta$ from $R$. Then, we have $||R+\Delta||_F \le ||R||_F + ||\Delta||_F$, due to triangle inequality. In other words, the Frobenius norm based score satisfies Lipschitz property, since
$$\frac{||R+\Delta||_F - ||R||_F}{||\Delta||_F} \le 1.$$
  Lipschitz property is a particularly important since there is a bound to the change in score, even though the auditor exhibits dynamic preferences regarding fairness notions.

- **Equal Treatment of Fairness Notions:** Frobenius norm of the matrix $R$ can also be represented as follows.
$$||R||_F = \sqrt{\text{Tr}(R^T R)} \tag{28}$$
  Let $R$, $UR$, and $RV$ be the reputation matrices of three different auditors. Then, we have
$$||UR||_F = ||RV||_F = \sqrt{\text{Tr}(R^T R)} = ||R||_F. \tag{29}$$
  In other words, the three auditors with reputation matrices which differ by orthogonal transformations have the reputation score. For the sake of illustration, consider two auditors: (i) one who complies with statistical parity, but not with calibration, and (ii) another who satisfies calibration but not statistical parity. Assuming that both (i) and (ii) treats all the remaining fairness notions identically, the Frobenius norm of their reputation matrices would be same. In other words, our reputation score treats all fairness notions equally.

# 7 Evaluation Methodology

In this section, we discuss different methodologies used to evaluated our proposed $\epsilon$-LAM based on simulation as well as real human audit data.

## 7.1 Datasets

We validate our theoretical findings using the following datasets, each of which are pre-processed as follows:

1. **ProPublica's COMPAS dataset (Larson et al., 2016):** In this paper, we perform same preprocessing procedure as done by ProPublica in their analysis in (Larson et al., 2016). The races in the dataset are only restricted to African-American, Caucasian, and other. We consider *females* and *Caucasians* as privileged groups (Bellamy et al., 2019). We consider the binary attribute called *two-year recidivism* (with labels being *most likely* or *least likely*) within the dataset as the output label in our analysis. If the *two-year recidivism* attribute is *least likely*, we deem the outcome as a favorable one. Furthermore, we discretize *age* into three age-groups (i.e. $< 25$, 25-45, $> 45$) and relabel this feature as *age category*. Similarly, *prior count* feature is also discretized into three discrete bins (i.e. 0, 1-3, $> 3$) as well.

2. **German credit data (Merz & Murphy, 1996):** In this dataset, we consider credit history, savings, employment, sex, and age as input features. Moreover, we categorize *age* into two groups: young ($< 26$) and old ($>=26$). We assume that males and older individuals as privileged groups and *credit risk* value of 1 (i.e. good credit risk) as the favourable outcome.

3. **Adult income dataset (Kohavi & Becker, 1994):** The objective is to predict whether the income of an individual is $> \$50K$ or $< \$50K$. The input features include age, sex, race, and education. In the pre-processing phase, the continuous feature *age* is transformed into different groups of ages (0-10, 11-20, and so on). For the feature *race*, we limited the labels to binary by mapping 'White' to 1 and all other races to 0. We have 32561 data tuples in total.

4. **Real human feedback data (Dressel & Farid, 2018):** This data acquisition experiment consists of a short description of the defendant (gender, age, race, and previous criminal history) is provided to the human auditor. A total of 1000 defendant descriptions are used that are drawn randomly from the original ProPublica's COMPAS dataset. Furthermore, these descriptions were divided into 20 subsets of 50 each. The experiment consisted of 400 different crowd workers and each one of them was randomly assigned to see one of these 20 subsets. The participants predicted whether a particular individual would recidivate within 2 years of their most recent crime. The original data consists whether a crowd worker predicted correctly or not compared to the original classification in the COMPAS dataset. We preprocessed this dataset and obtained the true prediction given by the crowd workers.

## 7.2 Evaluation of Comparative and Noncomparative Fairness Notions from Data

Comparative fairness notions provide a benchmark for fairness evaluation in today's ML-based systems. The evaluation of these comparative fairness notions, be it group fairness or individual fairness, have been studied extensively in the literature. This paper uses state-of-the-art approaches to evaluate comparative fairness of both ML-based system as well as the auditor in our simulation experiments and our analysis on real datasets. Furthermore, non-comparative fairness is also evaluated using the proposed LAM on the same datasets, and is compared with the empirical comparative fairness evaluations as demonstrated in Section 5.

The evaluation of individual fairness notion relies on a similarity metric $\mathcal{D}$ that quantifies the (dis)similarity between input profiles representing different individuals. The exact choice of similarity metric $\mathcal{D}$ is highly contextual to the ML-based system and the environment in which it is deployed. However, in a complex application setting, it is impractical to characterize $\mathcal{D}$ formally due to limited knowledge about the system and its environment. Nonetheless, one can learn the similarity metric $\mathcal{D}$ from observed participants in this system. Inspired from prior work (Zemel et al., 2013; Lahoti et al., 2019; John et al., 2020), we adopt an unsupervised learning algorithm based on clustering, where we construct context-dependent clusters of similar individuals using non-sensitive attribute features from respective datasets. For illustrative purposes,

| Sex | Age | Race | Prior Offenses | Charge Degree |
|---|---|---|---|---|
| Female | 25-45 | Caucasian | 1 to 3 | Misdemeanor |
| Male | 25-45 | Other | 0 | Felony |
| Female | Greater than 45 | African-American | 0 | Felony |
| Male | 25 - 45 | Other | More than 3 | Misdemeanor |
| Male | Greater than 45 | Other | 1 to 3 | Misdemeanor |

Table 1: Example of 5 different clusters present in COMPAS dataset

Table 1 presents 5 different clusters based on various attributes available within the COMPAS dataset. A family of similarity metrics based on Mahalanobis distance $\mathcal{D}(C) = \sqrt{(\boldsymbol{x}_i - \boldsymbol{x}_j)^T C^{-1} (\boldsymbol{x}_i - \boldsymbol{x}_j)}$ is used, where $\boldsymbol{x}_i, \boldsymbol{x}_j$ are input profiles in a dataset. The parameter $C$ is a positive semi-definite covariance matrix that is learned from data. For COMPAS, the features *juvenile counts* (both misdemeanor and felony counts), *prior offences* and *charge degree* are used to construct clusters based on estimated Mahalanobis distance. Similarly, for German credit dataset, we consider the following features *credit history*, *employment* and *savings*. Lastly, for the Adult income dataset, we use *education years* and *age* to measure similarity. A cluster is labeled as being violated by a given entity (be it the ML-based system, or the auditor) if all output labels are not identical. Therefore, we say an entity is individually fair if it has no cluster violations. Furthermore, in binary classification settings, the number of mismatches between the output labels at the ML-based system and the auditor can be evaluated within each cluster. We say that the auditor is $\epsilon$-LAM with respect to the ML-based system (i.e., a measure for non-comparative fairness evaluation) if there exists at most $\epsilon$ input profiles across all clusters, whose outcome labels at both the ML-based system and the auditor mismatches.

Group fairness notions rely on the difference between the probability of favorable outcomes at the unprivileged group to that of the privileged group. This paper evaluates the group fairness notions for the digital twins of both ML-based system ($\hat{g}_n$) and the auditor ($\hat{f}_n$) which are designed using standard ML algorithms - logistic regression, random forest, and support vector machine (SVM). However, in the context of equalized odds and calibration, the probabilities are evaluated conditioned on the true outcome labels. This paper assumes that these true labels are the original outcomes from the ML-based system ($g$) and the human auditor ($f$) respectively that are available from the simulated entities or real datasets.

### 7.3 Simulated Auditor Models

Real datasets are typically small in size ranging around 50 feedback samples per auditor, collected from about 100-500 auditors in total. Such data collection exercises are extremely expensive and are not always available in a timely manner. As a result, validation on real data does not provide a comprehensive evaluation of the proposed methodology. In order to address this issue, this paper also considers three simulated auditors based on decision trees to mimic human auditor's behavior on COMPAS, German credit, and Adult income datasets respectively. Although these decision trees are manually chosen by the authors and cannot be guaranteed to exist in real-world, their designs are inspired from relevant and contextual features that are not sensitive in terms of any protected attributes.

For COMPAS, the decision rule $s_1$ is defined below, using three input features, namely *number of prior offences* and *degree of the offence* (felony or misdemeanor), and a binary output label based on the attribute *two year recidivism* (most likely or least likely):

$$s_1(x, y, z) = \begin{cases} 0, & \text{if } x.\text{priors-count} \in [1, 3] \text{ and } x.\text{charge-degree} = \text{Felony} \\ & \qquad\qquad \text{OR} \\ & \text{if } x.\text{priors-count} > 3 \text{ and } x.\text{charge-degree} = \text{Misdemeanor} \\ 1, & \text{otherwise.} \end{cases} \tag{30}$$

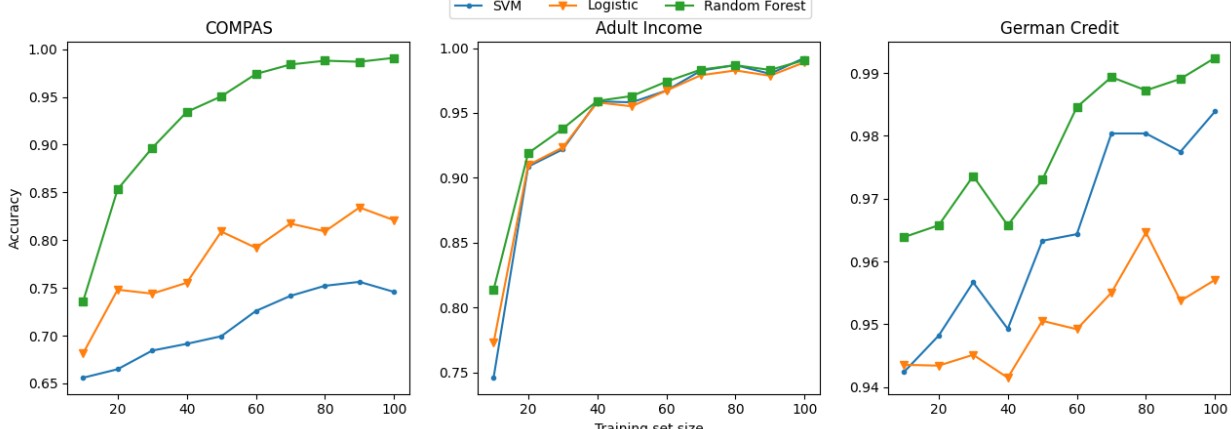

Figure 5: Accuracy of respective learning models while predicting the simulated auditor responses across different training set sizes.

Since the task of the Adult income dataset is to predict whether yearly income of an individual is $> 50K$ or $<= 50K$, the decision rule $s_2$ is constructed based on the feature *education* as shown below:

$$s_2(x, y, z) = \begin{cases} 0, & \text{if } x.\text{education} \in [\text{Bachelors, Masters, School Professor, Doctorate}] \\ 1, & \text{otherwise.} \end{cases} \tag{31}$$

Similarly, for German credit dataset, the decision rule $s_3$ is defined based on input features *savings, credit history* and *employment* are considered while designing the auditor's relation.

$$s_3(x, y, z) = \begin{cases} 0, & \text{if } x.\text{savings} > 500, \ x.\text{credit-history} = \text{Paid, and } x.\text{employment} > 2 \text{ years} \\ 1, & \text{otherwise.} \end{cases} \tag{32}$$

## 8 Experimental Results and Discussion

This section evaluates the proposed fairness evaluation methodology using three different experiments. Firstly, standard learning algorithms (e.g. logistic regression, support vector machine, and random forest) trained to mimic the auditor responses are evaluated in terms of learning accuracy and absolute error $|\mu - \hat{\mu}|$ in Section 8.1. Secondly, the digital twins of both the ML-based system and the auditor designed based on standard learning algorithms are evaluated in terms of both comparative and non-comparative fairness notions in Section 8.2. Lastly, the empirical distribution of the Frobenius norm based auditor reputation score is depicted and compared with other scoring functions on real human feedback data in Section 8.3.

### 8.1 Auditor Twin Evaluation

Auditor twins are evaluated in two ways, namely learning performance in terms of accuracy and fairness evaluation error in terms of absolute error $|\mu - \hat{\mu}|$, on the four datasets listed in Section 7.1 in the context of binary classification. While the labels $y = g(x)$, $z = f(x)$ and $s(x, y, z)$ are available in simulation experiments, the real human feedback data (Dressel & Farid, 2018) does not contain the latent subjective label $z = f(x)$ explicitly. However, as stated in Remark 1, the auditor's latent subjective label $z$ can be inferred from the system's true label $y$ and elicited feedback $s$ using the XOR relation $z = y \oplus s$ in binary classification settings. In each experiment, 25 random train-test splits are performed on all four datasets and average performance metrics are used to gain insights into the performance evaluation of the proposed auditor twin algorithms.

*Simulated Auditor on COMPAS, Adult and German Credit Datasets:* The training set size is varied from 10 to 100 data samples and the accuracy in predicting the simulated auditor's responses is recorded for

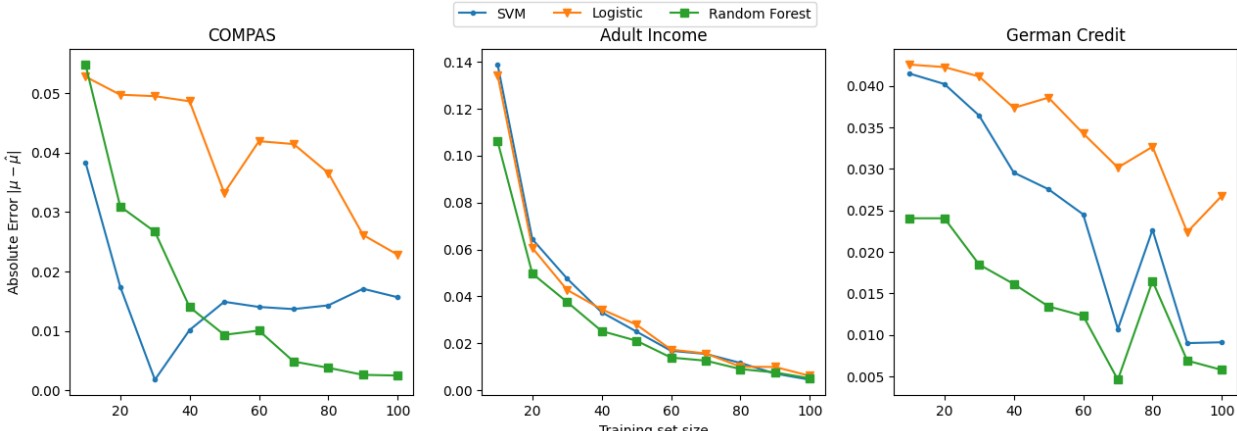

Figure 6: Absolute error between DEFM and IEFM across varied training set sizes.

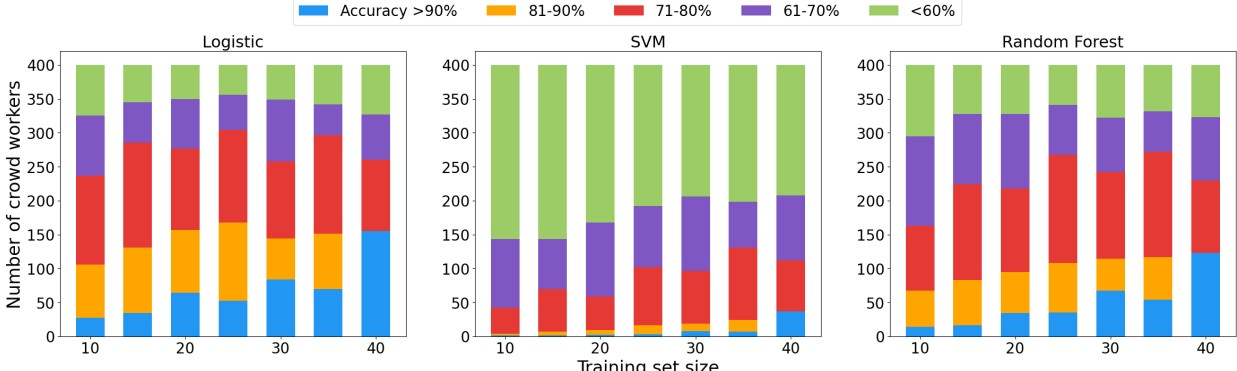

Figure 7: Number of auditor twins achieving a prescribed accuracy as a function of training set size, for different learning models.

each training size. Figure 5 shows that the accuracy in predicting $\hat{z} = \hat{f}(x)$ increases with training set size. More specifically, the random forest algorithm outperforms both logistic regression and SVM in terms of both accuracy and absolute error $|\mu - \hat{\mu}|$. This is because the three simulated auditors $s_1$, $s_2$ and $s_3$ resemble that of a decision tree which makes random forest algorithm into a realizable learning framework. In addition, the absolute error between DEFM and IEFM (ref. to Figure 6) is close to zero across all three learning algorithms. This signifies that auditor's responses can be efficiently learned even with small number of samples. Note that evaluators typically incur a very high price for eliciting a large number of feedback samples. For all practical purposes, since eliciting more than 100 feedback samples per auditor is infeasible, the x-axis in Figures 5 and 6 is limited to at most 100 training samples.

*Real Human Feedback Data:* Similar to the earlier experiment, the accuracies in predicting real human responses is recorded for different training set sizes varying from 10 to 40 samples. Figure 7 shows that the number of crowd auditors whose twins' responses have an accuracy of at least 90% (blue bars) increases with the training set size across all three learning algorithms. Although random forest outperformed other learning algorithms in reliably mimicking the simulated auditors, logistic regression outperforms random forest based twin in real human feedback data. However, SVM recorded low accuracy performance in a majority of real human auditors. Figure 8 depicts the histogram of the absolute error $|\mu - \hat{\mu}|$ observed on real human feedback data, as well as the best fitting gamma distribution that minimizes the negative log-likelihood function (for more details, refer to Virtanen et al. (2020)). Regardless of the learning algorithm, Figure 8 illustrates the absolute error between DEFM and IEFM to be smaller than 0.2 with high probability amongst real crowd auditors.

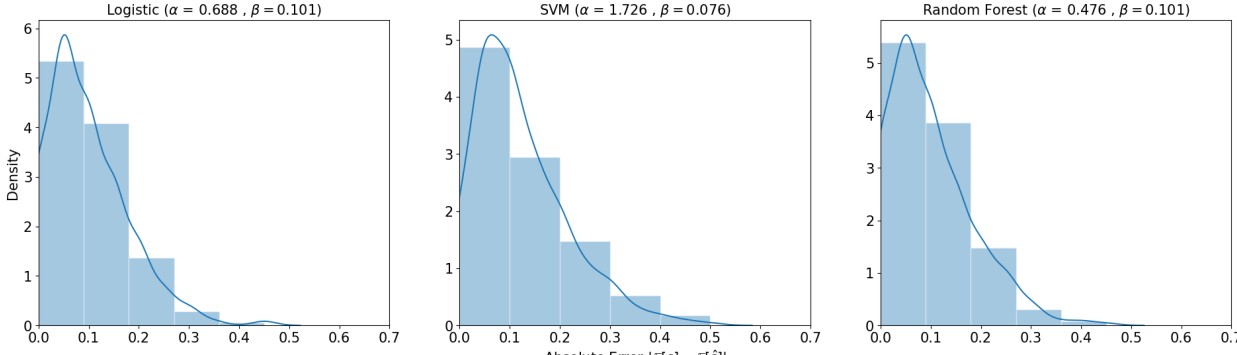

Figure 8: Distribution of crowd auditors in terms of error between DEFM and IEFM.

| Dataset | # Test Samples | Total # Clusters | # Clusters Violated by System | # Clusters Violated by Auditor | # LAM-based Mismatches |
|---|---|---|---|---|---|
| COMPAS | 1543 | 70 | 30 | 0 | 57 |
| German credit | 250 | 25 | 21 | 0 | 25 |
| Adult income | 8141 | 62 | 55 | 0 | 59 |

Table 2: Comparison between individual fairness evaluations at system and auditor, and their LAM mismatches for three simulated auditor twins across various datasets.

## 8.2 Demonstrating the Relationship between Non-Comparative and Comparative Notions

In this section, the digital twins of both ML-based system and the auditor are evaluated in terms of comparative and non-comparative fairness notions. In terms of comparative fairness, both individual fairness and group fairness are evaluated as per the methodology discussed in Section 7.2. The labels of both the ML-based system and the auditor are learned using standard learning algorithms (e.g. logistic regression, SVM, and random forest). The results are averaged across 25 random train-test splits.

*Simulated Auditor on COMPAS, Adult and German Credit Datasets:* The training set size is considered as 50 samples, since it incurs a very high price for eliciting a large number of samples. The respective entity's (the ML-based system or the auditor) biases are evaluated on remaining test set predictions. In terms of *individual fairness*, Table 2 shows that the simulated auditor is individually fair (0 violated clusters). However, the auditor does not satisfy $\epsilon$-LAM with respect to all three ML-based systems due to a large number of LAM-based output label mismatches between the twins of the simulated auditor $\hat{f}_n(x)$ and that of the system $\hat{g}(x)$ shown in Table 2. In other words, even though the simulated auditor is individually fair, the system is not individually fair in practice, if the auditor is not $\epsilon$-LAM with respect to the system. However, the fairness of the simulated auditor is heteroskedastic in terms of *group fairness* notions. We measure the simulated auditor's biases with respect to three group fairness notions (statistical parity, equal opportunity and calibration) based on different sensitive attributes (gender and race/age). Based on our insights from Section 8.1, the fairness evaluations due to random forest classifer are treated as a benchmark for auditor performance. Note that both COMPAS and German credit datasets exhibit biases in fairness evaluation in both logistic regression as well as SVM models. However, as observed in Figure 9b, the Adult income dataset seems to be robust to ML models in terms of any group fairness notion since all ML models have similar learning performance, as shown in Figure 5. Such behavior is consistent with Theorem 2. Furthermore, note that if the auditor is fair with respect to any group fairness notion (e.g. refer to equalized opportunity value for the auditor's twin in COMPAS dataset in Figure 9a), the system is not fair if the auditor's twin is not $\epsilon$-LAM with respect to the system's twin.

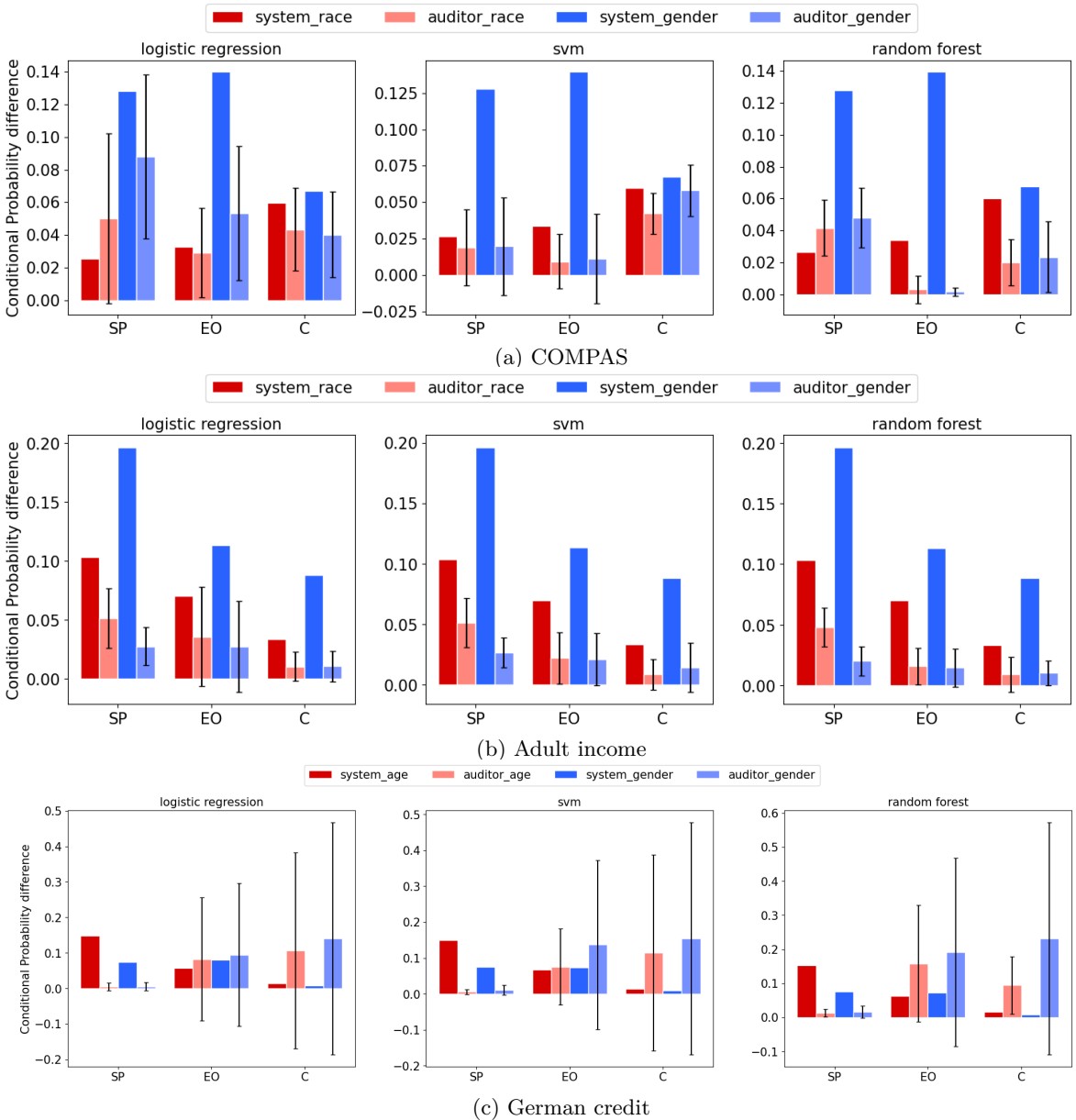

Figure 9: Evaluation of real-world classifiers and the simulated auditor twin for statistical parity (SP), equal opportunity (EO) and calibration (C), with respect to different protected attributes.

*Real Human Feedback Data:* Based on earlier results (Figure 7), the choice of ML algorithm varies from one auditor to another, as well as the application context. Therefore, we train each crowd auditor's responses using all three ML algorithms (logistic regression, SVM, and random forest) and select the one which yields in highest accuracy in predicting their responses. Specifically, logistic regression predicted 57.5% of crowd auditors' responses with high accuracy. Whereas, SVM and random forest classifiers predicted 25% and 17.5% of crowd auditor's responses with high accuracy respectively. In terms of *individual fairness*, Figure 10 shows that only 20 auditors (in a total of 400 auditors) are individually fair (0 violated clusters). The majority of the crowd auditors violate 5% to 20% of the clusters present in the subset given to them. Figure 10 presents the best-fit gamma distribution of individually fair auditors whose parameters are estimated as $\alpha = 12.75$ and $\beta = 0.0187$. Furthermore, we evaluate the crowd auditors by varying the threshold $\delta$ in the definition of *group fairness* notions (statistical parity, equalized odds, calibration, and equal accuracy), as stated in Section 2.1. Each crowd auditor's performance is assessed based on the four group fairness notions

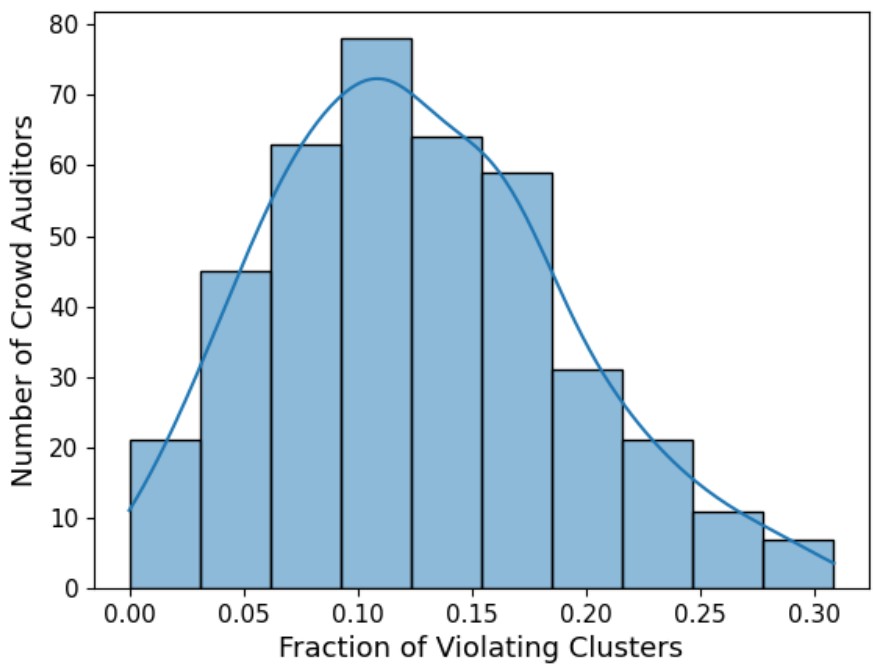

Figure 10: The distribution of crowd auditors as a function of fraction of clusters violating individual fairness.

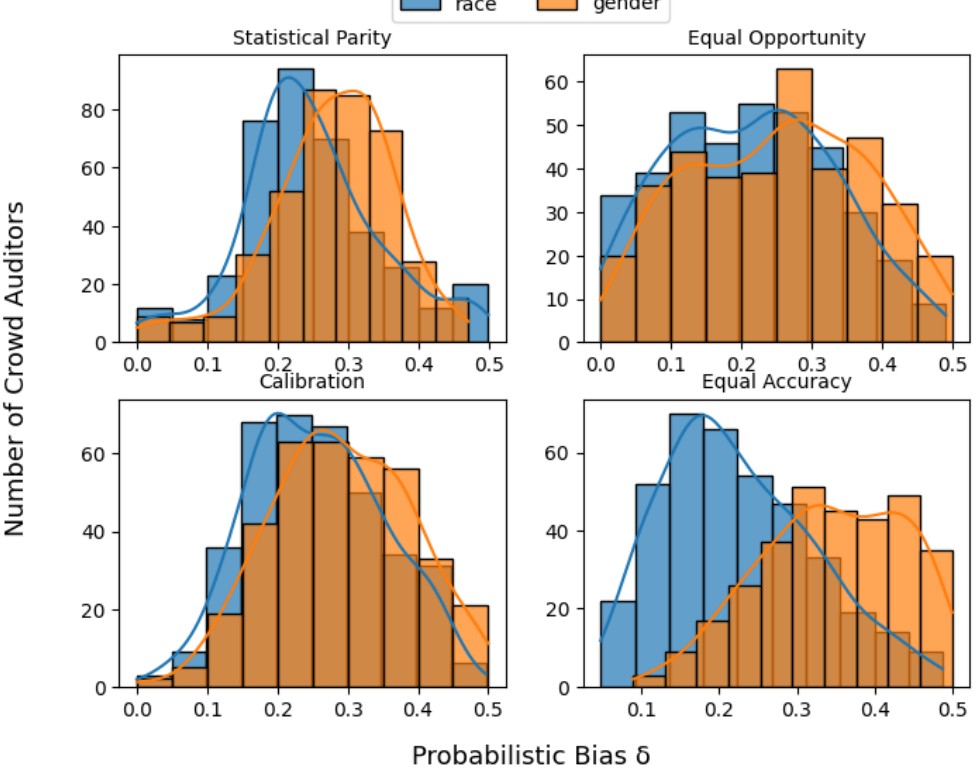

Figure 11: Number of crowd auditors satisfying different group fairness notions across varied probabilistic bias with respect to race and gender.

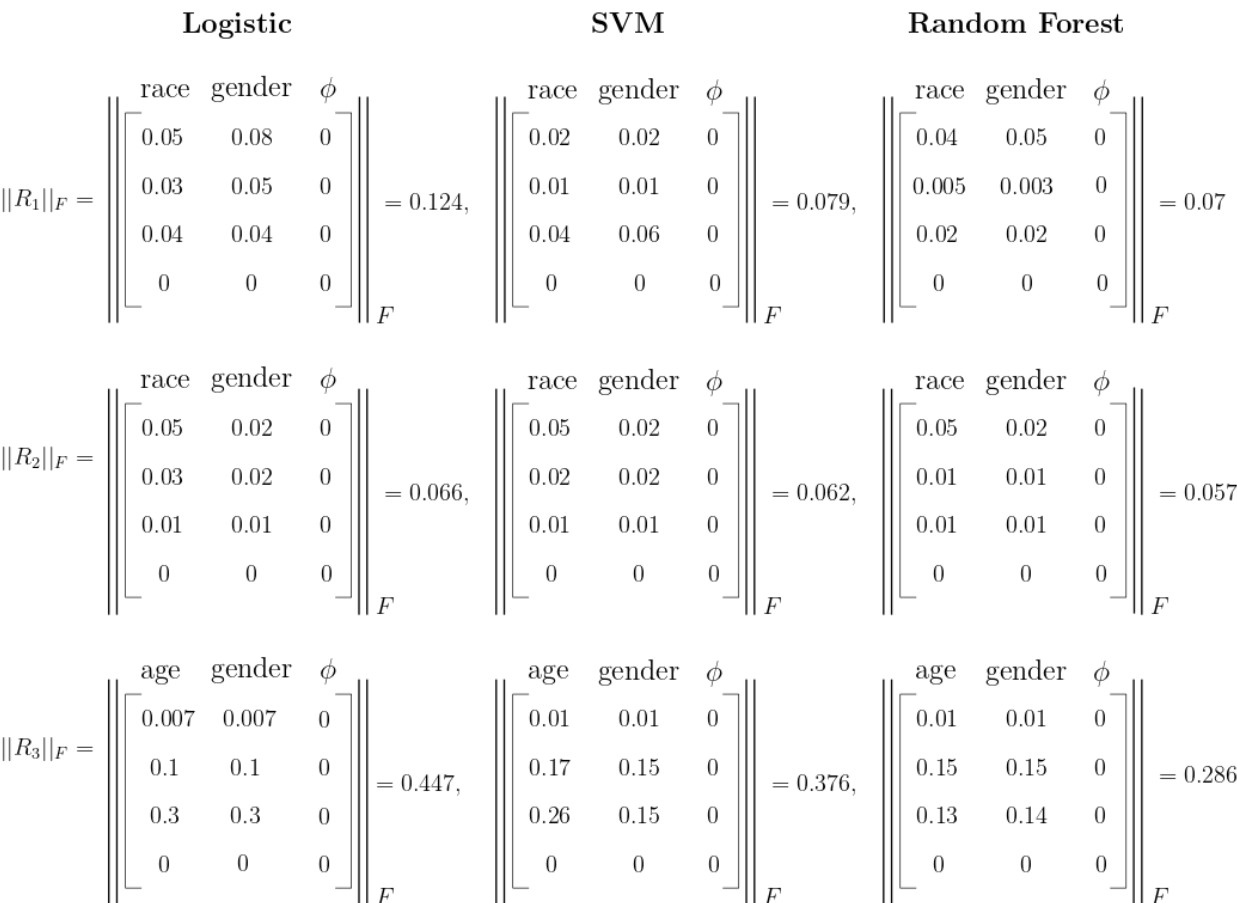

Figure 12: Reputation Matrices for the Three Simulated Auditors $s_1$, $s_2$ and $s_3$ on COMPAS, Adult Income and German Credit Datasets respectively

by varying the threshold $\delta$ from 0 to 0.5. Figure 11 shows that crowd auditors are typically more fair with respect to race than gender based on the shift in distribution.

### 8.3  Reputation Scores Interpretation and its Relation with Learning Performance

In this section, auditor reputation scores are interpreted using simulation experiments and then evaluated on real experiments. The results are discussed below in the following two subsections respectively. Note that lower reputation scores correspond to lower biases, which can be attributed to higher audit reliability.

*Simulated Auditor on COMPAS, Adult and German Credit Datasets:* As discussed in simulation results in Section 8.2, random forest classifier is considered the benchmark in this analysis. Figure 12 shows reputation matrices of the simulated auditors on COMPAS, Adult income and German credit across three learning algorithms. The rows in the reputation matrix $R$ represent different comparative fairness notions in the following top-down order: statistical parity, equal opportunity, calibration, and individual fairness. Whereas, the columns represent different sensitive attributes as shown in Figure 12. Note that the null attribute $\phi$ is included in the attribute-set since individual fairness is agnostic to sensitive attributes. The reputation scores of all twins for the Adult dataset is similar due to their identical learning performance. However, this is not the same for other datasets. For instance, random forest classifier outperforms logistic regression and SVM in COMPAS and German credit datasets, even in terms of their reputation scores as well. Another interesting observation to note is that learning errors do not increase biases identically across all fairness notions. For example, in the case of $R_3$, statistical parity and equal opportunity notions are similar, but the logistic regression model registered a significant chance in calibration in logistic regression and SVM models. This is automatically reflected in their reputation scores as well.

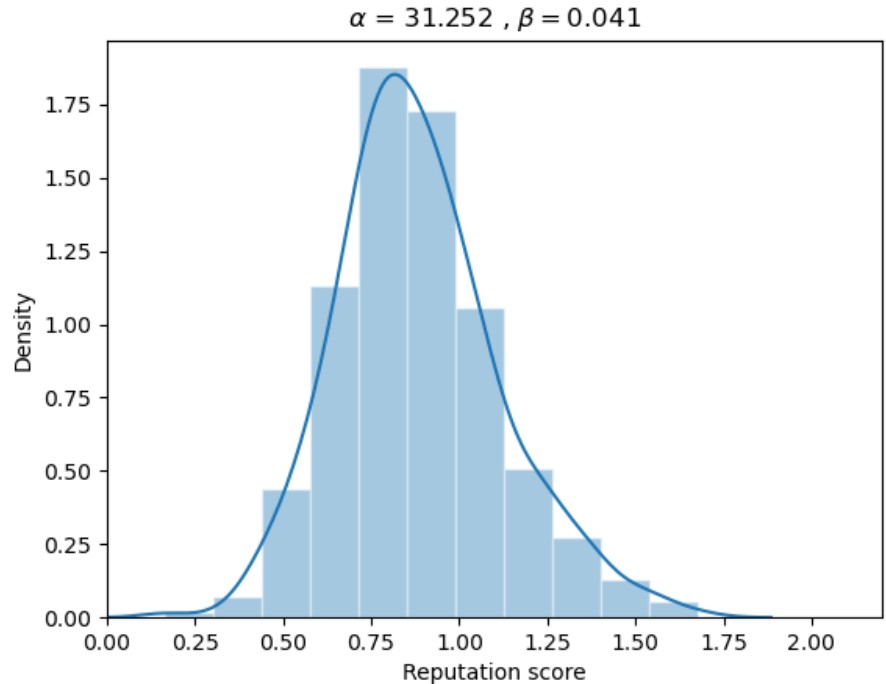

Figure 13: The Frobenius reputation score distribution of 400 crowd workers.

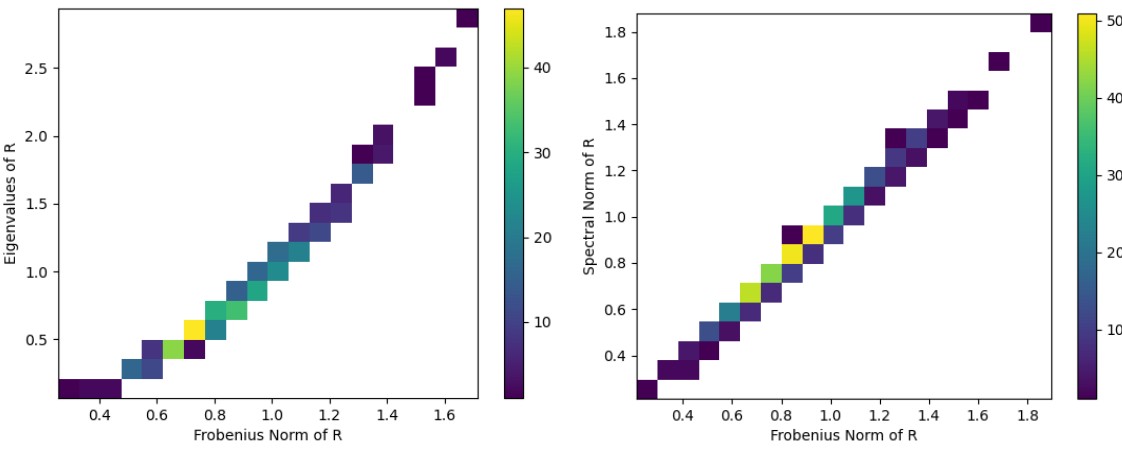

(a) Distribution of crowd auditors' Frobenius reputation vs. Eigenvalues reputation

(b) Distribution of crowd auditors' Frobenius norm reputation vs. spectral norm reputation

Figure 14: A comparison of different reputation scoring functions

*Real Human Feedback Data:* The histogram of reputation scores based on real data elicited from 400 crowd workers is plotted in Figure 13. A best-fit gamma distribution is also evaluated via minimizing the negative log-likelihood function, whose parameters were found to be $\alpha = 31.25$ and $\beta = 0.04$. Note that the reputation scores range between 0.16 and 1.67, with a significant majority of them lying between 0.5 and 1.0. Furthermore, we compare the proposed Frobenius norm reputation of the auditor with two different scoring functions i.e., eigenvalues and spectral norm of the reputation matrix $R$. On the final note, Figure 14 depicts a monotonic relationship between Frobenius norm based reputation score and eigenvalue/spectral-norm based scoring mechanisms. However, the other scoring mechanisms do not necessarily follow the axiomatic properties of fairness as stated in Section 6.

## 9 Conclusion and Future Work

We developed a novel latent assessment model to characterize human auditor feedback and demonstrated its relationship with traditional fairness notions both theoretically and on real datasets. We obtained PAC learning guarantees on learning auditor's intrinsic fairness assessments, and demonstrated the learning performance of three learning algorithms on a real human feedback dataset. Consequently, this paper enabled us to accomplish two important challenges in the design of a crowd-auditing platform: (i) we can learn/mimic auditor's intrinsic evaluations using little elicited feedback and automate the evaluation on the remaining possibilities especially in high-dimensional learning algorithms, and (ii) we can also evaluate auditor biases with respect to diverse traditional fairness notions. In addition, we use the relationship between LAM and traditional fairness notions to identify reliable auditors for feedback elicitation based on their reputation scores.

In future, we will address all the other challenges in the design of crowd-auditing platforms. Since feedback elicitation is an expensive process, we will improve our LAM model to account for feedback for data bundles, as opposed to our current feedback model for singleton data tuples. Furthermore, we will also investigate appropriate fusion rules to aggregate feedback collected from multiple auditor with heterogeneous opinions based on their reputation.

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
