# OpenReview forum: "Auditor Fairness Evaluation via Learning Latent Assessment Models from Elicited Human Feedback"
_TMLR — Rejected by TMLR_

### Review · Reviewer_MQ16 · 2022-06-13

**Summary Of Contributions:**

The paper proposes the so-called epsilon-Latent Assessment Model, in order to characterize binary feedback from human auditors. The authors study various links between this notion and existing individual and group notions of fairness. They provide a way to measure the overall bias of a human auditor. They also provide experiments on standard fair ML datasets to demonstrate their points.

**Broader Impact Concerns:**

None that I can think of.

**Requested Changes:**

In order for the paper to be potentially ready for acceptance, I believe that the authors should address the issues raised above, in particular:

- Clarify what the contributions of the paper are and how they address the limitations of prior work.
- Explain why and how asking auditors to evaluate the accuracy of a model is related to auditing fairness.
- Improve on Section 4, making the assumptions formal and statements more precise/easy to follow.

Minor points:

- The authors may want to consider swapping Sections 1.1 and 1.2 to improve readability.
- In equation (6), it will be good to state explicitly that s depends on x and y.


**Strengths And Weaknesses:**

Strengths: The paper studies the important problem of leveraging human feedback in order to access the fairness of ML models and more generally to understand what notions are fairness are important in machine learning.

Weaknesses: Unfortunately, I see several important limitations of the paper, described below:

- The paper is motivated by several limitations of existing fairness measures, in particular that they are context-dependent and that they are often conflicting, so they can not be satisfied at the same time. I agree with these limitations, but I do not see how the paper addresses them. The proposed framework rather deals with collecting feedback from human auditors on whether they agree with the decisions of a classifier and on then evaluating the biases of the human auditors and the extend to which they exhibit various forms of fairness in their own decisions. Unfortunately, I do not see what in the methodology and the experiments actually helps to suggest what fairness notion is appropriate in a given context or how to choose a trade-off between different fairness measures for a given task.

- By asking human auditors to state whether they agree with a classification decision, which is the essence of the proposed epsilon-LAM framework, one essentially asks the auditors whether an existing classifier is accurate, not whether it is fair. Many notions of fairness are orthogonal to accuracy and therefore I don't think that the feedback that would be received in such a model is fair/unfair, but rather accurate/inaccurate.

In a sense, the proposed method is closer to the problem of evaluating the quality of crowdsourcing workers - a comparison to these methods is present only is Section 5.3, it will be good to discuss connections to this literature more extensively.

- Regarding the theoretical guarantees, I find that the mathematical assumptions behind the results are not clearly stated and therefore it is hard to judge their validity. For example, many of the results require g to be epsilon-LAM with respect to f. This concept is undefined - we only know what it means to for an auditor to satisfy epsilon-LAM. In Proposition 3, the authors assume that "the probability distributions are M-Lipschitz continuous over all possible f and g functions". It is not clear which distributions are in question. In addition (Lipschitz-)continuity is usually defined with respect to a set of parameters and a distance measure, so again it is not clear what the authors mean. Finally, I have to admit that I am completely lost in the statement of Theorem 1.

---

> ### Author Response · Authors · 2022-08-09
> **Our Responses to Reviewer MQ16 (Part 1)**
>
> *We sincerely thank the reviewer for valuable feedback. However, the feedback has proved that our work needs major changes. Unfortunately, we were unable complete all the major changes in the given time. We request the reviewer to give two-week extension to complete all the concerns and submit the revised version.*
>
> **Our response to Reviewer MQ16's comments and the corresponding changes in the original paper will be highlighted in RED color.**
>
> **Strength 1**
> The paper studies the important problem of leveraging human feedback in order to access the fairness of ML models and more generally to understand what notions are fairness are important in machine learning.
>
> **Response**
> We thank the reviewer for the feedback and sincerely appreciate it.
>
> **Weakness 1**
> The paper is motivated by several limitations of existing fairness measures, in particular that they are context-dependent and that they are often conflicting, so they can not be satisfied at the same time. I agree with these limitations, but I do not see how the paper addresses them. The proposed framework rather deals with collecting feedback from human auditors on whether they agree with the decisions of a classifier and on then evaluating the biases of the human auditors and the extend to which they exhibit various forms of fairness in their own decisions. Unfortunately, I do not see what in the methodology and the experiments actually helps to suggest what fairness notion is appropriate in a given context or how to choose a trade-off between different fairness measures for a given task.
>
> **Response 1**
> We sincerely apologize for the ambiguity in articulating our paper's contributions. Our goal was not to propose a new fairness notion. Instead, we identify that most ML algorithms are designed for social applications where true labels do not exist, but people make subjective judgements that are usually contaminated with personal biases. To address this concern, we introduce a new idea of justice called noncomparative fairness, which questions the fairness judgements made by auditors according to the treatment/service provided to everyone based on their own features/merit. We propose a novel latent assessment model to characterize auditor's non-comparative judgements, and develop a scalable approach to audit high-dimensional machine learning systems via learning the model on small datasets elicited from real human auditors, and automating it to all other possibilities for a comprehensive fairness evaluation. We also present a novel reputation metric to evaluate the auditor's biases from both noncomparative and comparative fairness perspective. In an attempt to clarify these contributions, we have rewritten the entire introduction section from this perspective.
>
> **Weakness 2**
> By asking human auditors to state whether they agree with a classification decision, which is the essence of the proposed epsilon-LAM framework, one essentially asks the auditors whether an existing classifier is accurate, not whether it is fair. Many notions of fairness are orthogonal to accuracy and therefore I don't think that the feedback that would be received in such a model is fair/unfair, but rather accurate/inaccurate.
>
> **Response 2**
> We agree that our latent assessment model resembles accuracy, a metric used traditionally to evaluate machine learning algorithms. It is important to note that accuracy makes sense when true labels are available for comparison with predictions. However, in most practical social applications (e.g. criminal justice, banking, job markets), there is no one true label since labelers offer non-identical responses based on diverse merit-based criteria. Therefore, it is insufficient to investigate fairness in ML systems from a comparative perspective. Our paper rekindles a new notion called non-comparative fairness which was originally found in the justice literature, which focuses on how fairly people are treated based on their own merit, without comparing them with others. In summary, our paper addresses the gaps that arise due to the insufficiency of accuracy as a performance criteria in evaluating machine learning algorithms. We articulate this in our updated introduction section.

---

> ### Author Response · Authors · 2022-08-09
> **Our Responses to Reviewer MQ16 (Part 2)**
>
> **Weakness 3**
> In a sense, the proposed method is closer to the problem of evaluating the quality of crowdsourcing workers - a a comparison to these methods is present only is Section 5.3, it will be good to discuss connections to this literature more extensively.
>
> **Response 3**
> As suggested, we have looked into crowdsourcing literature and found many papers on measuring worker quality. However, most papers rely on finding true labels (e.g. as in the case of truth discovery algorithms), which are not relevant to social applications such as criminal justice and banking. Furthermore, the reputation of an auditor needs to be evaluated based on both non-comparative and comparative fairness notions, which was never investigated in the past to the best of our knowledge. We clarify this in the updated discussion on crowd workers' quality in Related Work section.
>
> **Weakness 4**
> Regarding the theoretical guarantees, I find that the mathematical assumptions behind the results are not clearly stated and therefore it is hard to judge their validity. For example, many of the results require g to be epsilon-LAM with respect to f. This concept is undefined - we only know what it means to for an auditor to satisfy epsilon-LAM.
>
> **Response 4**
> As suggested, we updated the proposition statements to be consistent with the definition of LAM.
>
> **Weakness 5**
> In Proposition 3, the authors assume that "the probability distributions are M-Lipschitz continuous over all possible f and g functions". It is not clear which distributions are in question. In addition (Lipschitz-)continuity is usually defined with respect to a set of parameters and a distance measure, so again it is not clear what the authors mean.
>
> **Response 5**
> The Lipschitz continuity of probability distributions of the auditor $f$ and the system $g$ are defined based on the input parameter $x$ which defines the attributes of the person being served, and the distance function is the similar metric used to evaluate the similarity of any two individuals in the population $\mathcal{X}$. We clarified this confusion in the paragrph before Proposition 3.
>
> **Weakness 6**
> Finally, I have to admit that I am completely lost in the statement of Theorem 1.
>
> **Response 6**
> We clarified this issue by articulating the need to learn the LAM model and mimic the auditor. A auditor twin enables us to evaluate high-dimensional machine learning systems which are often found in social applications due to the presence of a large number of diverse individuals, which cannot be otherwise evaluated due to significant costs incurred during a comprehensive data collection task. Theorem 1 specifically provides the minimum number of samples that we need to collect from a given auditor to successfully mimic the same within prescribed learning guarantees. Such a result enables us to plan the task of feedback elicitation from the auditors.
>
> **Requested Changes**
> Clarify what the contributions of the paper are and how they address the limitations of prior work. -- Done.
>
> Explain why and how asking auditors to evaluate the accuracy of a model is related to auditing fairness. -- Done.
>
> Improve on Section 4, making the assumptions formal and statements more precise/easy to follow. -- Done.
>
> **Minor points**
> The authors may want to consider swapping Sections 1.1 and 1.2 to improve readability. -- Done.
>
> In equation (6), it will be good to state explicitly that s depends on x and y. -- Done.

---

### Review · Reviewer_Syg5 · 2022-06-14

**Summary Of Contributions:**

The paper aims to study the problem of evaluating the fairness of an auditor who is evaluating some ML model. More specifically, it asks a human auditor a binary question of whether the prediction on a certain individual is "fair" or not — i.e. whether or not the prediction given by the model is very different from the prediction that the auditor would have given. Then, the authors propose to learn the auditor’s intrinsic rule f from such fairness feedback given by the auditor and denote this learned rule $\hat{f}$ as Auditor Twin in figure 2. Then, they measure various unfairness with respect to hat{f} on a bigger dataset for which the auditor’s feedback has not been provided by predicting the auditor's feedback with hat{f}. There are a couple of experiments performed on datasets such as COMPAS, German credit data, Adult Income, and so on.

**Broader Impact Concerns:**

I don’t see any further broader impact concerns.

**Requested Changes:**

More than anything, the paper needs to explain the fundamental benefit of using this LAM framework as opposed to directly measuring the unfairness of f and/or g directly.

For the experiment, it is insufficient to simply report the unfairness measured by imputing the auditor’s feedback on some bigger dataset with the learned hat{f}.
(1) f’s true unfairness:  Because we actually have the intrinsic rule f, f’s unfairness measured on this bigger dataset should be reported.
(2) f’s empirical unfairness: the epirical unfairness measured on the original training dataset {x_i, y_i, f(x_i)}_{i=1}^{n} should be reported.

It needs to be shown that the unfairness measured by imputing the auditor’s feedback on some bigger dataset with learned hat{f} is closer to (1) f’s true unfairness than (2) f’s empirical unfairness is to (1) in order to establish that LAM is better than directly measuring the unfairness of f and/or g directly.

**Strengths And Weaknesses:**

Frankly, it is hard to provide strengths and weakness of the paper as I don’t understand the fundamental contribution of the paper. So, I’ll use this section to describe my confusion: it is unclear whether the goal of the paper is to measure the unfairness of the ML model or the auditor that is evaluating the ML model with respect to some fairness notion. Either way, I do not exactly understand why the proposed LAM approach is needed.

If the goal is the former (i.e. measuring the unfairness of the ML model), it is unnecessary to go through an auditor and one can directly learn the unfairness (e.g. difference between positive rates) of the model from a sample. For instance, the empirical false positive difference rates difference should be concentrated around the true false positive rates difference. See generalization results from “Preventing Fairness Gerrymandering: Auditing and Learning for Subgroup Fairness” by Kearns et al to see generalization of group fairness and “Provably Approximately Metric-Fair Learning” by Rothblum and Yona for generalization of individual fairness. Most of these generalization results has the convergence rate of $1/\sqrt{n}$ where n the number of sample.

If the goal is the learn unfairness of an auditor with respect to some fairness notion, I don’t see what is the advantage of the proposed method over estimating it with the empirical unfairness directly. For instance, in the case of binary prediction, we exactly back out what prediction the auditor would have made for x as noted in the paper: fairness evaluation s is simply XOR of the model’s prediction $g(x)$ and the auditor’s prediction $f(x)$, so given $s$ and $g(x)$, one may back out $f(x)$. Also, more simply, we could have simply asked $f(x)$ from the auditor, which is still a binary question in this case. Then, given a dataset (x_i, y_i, f(x_i))_{i=1}^{n} where n is the total number of feedbacks we have gotten from the auditor, we can once again measure the empirical unfairness (e.g. false positive rate difference) with respect to f(x) from this dataset (x_i, y_i, f(x_i))_{i=1}^{n}. The convergence rate of unfairness estimated this way should be $1/\sqrt{n}$ as before.

The proposed method of this paper is to learn hat{f} from this dataset (x_i, y_i, f(x_i))_{i=1}^{n} and measure the unfairness on a separate dataset for which we don’t have the auditor’s feedback by imputing the auditor’s feedback with \hat{f}. Information theoretically speaking, measuring unfairness with respect to hat{f} on this bigger dataset cannot be better than the unfairness measure with respect to the original (x_i, y_i, f(x_i))_{i=1}^{n}.

Of course, collecting f(x) (i.e. asking the auditors for their opinion) is an expensive process, but there's no free lunch in that estimating f with some hat{f} and measuring the unfairness simply with respect to hat{f} is not a viable alternative as it will further propagate the error of hat{f} (i.e. how different hat{f} is from f).


Further confusions:
Section 4.2 shows what one can show if the auditor’s intrinsic rule f is eps-LAM with respect to the model g (i.e. f makes a similar prediction as the model g). More formally, they show that if f and g is similar in the sense of eps-LAM, g’s fairness with respect to some fairness notion implies f’s fairness and vice versa. As discussed above, I don’t immediately see how auditing for unfairness of f and/or g via this relationship between them is more helpful as compared to measuring the unfairness of f and/or g directly. Information theoretically speaking, estimating unfairness of f or g directly can only be better than measuring indirectly via this relationship I believe.


Also, there’s some discussion about learning the auditor’s prediction \hat{f} from the elicited feedback from the auditor in section 4.3. This learned \hat{f} is only useful if (1) the learned intrinsic rule hat{f} is actually close to f and (2) f is actually eps-LAM to g. However, whether or not the auditor’s intrinsic rule f is eps-LAM to g is not something that we can control, so I’m not sure exactly what theorem 1 is trying to convey.
Furthermore, g is a blackbox ML model we have access to so I don’t understand why there is a need to learn \hat{g} at all.
In addition, in theorem 1, it’s stated that it outputs hat{f} and hat{g} such that d(f(x), g(x)) <= epsilon with high probability. As explained before, note that whether or not auditor’s intrinsic rule f is eps-LAM with respect to the model g (how close f is to g) does not depend on the algorithm used to learn hat{f} and hat{g} and the number of samples we have to learn hat{f} and hat{g}. So, it cannot be that the algorithm always outputs d(f(x), g(x)) <= eps with high probability always. So, I take it that the intended theorem statement is that d(hat{f}(x), hat{g(x))<=epsilon. However, if the intrinsic rule f is very different from the ML model g (it is not eps-LAM), then I think it’s impossible to have both d(hat{f}(x), hat{g(x))<=epsilon and hat{f} is close to f.


Finally, PAC learnability is always stated with respect to some hypothesis class regardless of whether we are in a realizable setting or agnostic setting. It seems like there’s an inherent assumption that e are in a realizable setting for the theory part.

In terms of experiment, my understanding is that they have difference heuristics rules f for each dataset that the auditor may possibly use (as defined in page 10) and try to learn this rule f via different methods (logistic regression, SVM, and random forest) from some training dataset {x_i, y_i, f(x_i)}_{i=1}^{n} and measure the unfairness of this learned rule hat{f} on a bigger dataset. However, as I voiced the concern above, one needs to compare this to direct unfairness measurement from {x_i, y_i, f(x_i)}_{i=1}^{n}.

Finally, there needs to be more explanation about individual fairness definition used here. The individual definition proposed in this paper is slightly different than the original individual fairness definition which is |f(x) - f(x')| <= d(x,x') + slack where f is some soft classifier (f outputs some number from [0,1] instead of {0,1}).

---

> ### Author Response · Authors · 2022-08-09
> **Our Responses to Reviewer Syg5 (Part 1)**
>
> *We sincerely thank the reviewer for valuable feedback. However, the feedback has proved that our work needs major changes. Unfortunately, we were unable complete all the major changes in the given time. We request the reviewer to give two-week extension to complete all the concerns and submit the revised version.*
>
> Our response to Reviewer Syg5's comments and the corresponding changes in the original paper will be highlighted in BLUE color.
>
> **Weakness 1**
> Frankly, it is hard to provide strengths and weakness of the paper as I don’t understand the fundamental contribution of the paper. So, I’ll use this section to describe my confusion: it is unclear whether the goal of the paper is to measure the unfairness of the ML model or the auditor that is evaluating the ML model with respect to some fairness notion. Either way, I do not exactly understand why the proposed LAM approach is needed.
>
> **Response 1**
> We sincerely apologize for the ambiguity in articulating our paper's contributions. The same concern was also raised by the reviewer \textbf{MQ16}. So, we have addressed both the reviewers' concern together via rewriting the introduction section from a non-comparative fairness standpoint, as opposed to the comparative notions found in the algorithmic fairness literature. Our goal was to address the gaps present in current comparative fairness literature which rely on accuracy to evaluate ML algorithms. As a solution, we introduce a novel perspective of fairness called non-comparative fairness which treats every individual based on their merits. Specifically, we propose a latent assessment model to capture the auditor's non-comparative judgments. Furthermore, we evaluate the biases of the auditor using a novel reputation mechanism based on both comparative and non-comparative fairness notions.
>
> **Weakness 2**
> If the goal is the former (i.e. measuring the unfairness of the ML model), it is unnecessary to go through an auditor and one can directly learn the unfairness (e.g. difference between positive rates) of the model from a sample. For instance, the empirical false positive difference rates difference should be concentrated around the true false positive rates difference. See generalization results from “Preventing Fairness Gerrymandering: Auditing and Learning for Subgroup Fairness” by Kearns et al to see generalization of group fairness and “Provably Approximately Metric-Fair Learning” by Rothblum and Yona for generalization of individual fairness. Most of these generalization results has the convergence rate of  where n the number of sample.
>
> **Response 2**
> We thank the reviewer for suggesting some interesting papers. The work of Micheal Kerns et al. in "Preventing Fairness Gerrymandering:
> Auditing and Learning for Subgroup Fairness" explore the problem of auditing binary classifiers for false positive rates and statistical parity across infinitely many subgroups. On the other hand, "Provably Approximately Metric-Fair Learning” by Rothblum and Yona propose a relaxed version of individual fairness where, the fairness is guaranteed not just for training data but also for the underlying population distribution. Although these approaches provide convergence guarantees, they rely on availability of true labels which are often inaccessible in social applications such as criminal justice and banking. Furthermore, the two approaches explore biases from a comparative fairness perspective which do not question the fidelity of training data. However, our main goal was to measure the unfairness of the auditor and not the system's.

---

> ### Author Response · Authors · 2022-08-09
> **Our Responses to Reviewer Syg5 (Part 2)**
>
> **Weakness 3**
> If the goal is the learn unfairness of an auditor with respect to some fairness notion, I don’t see what is the advantage of the proposed method over estimating it with the empirical unfairness directly. For instance, in the case of binary prediction, we exactly back out what prediction the auditor would have made for x as noted in the paper: fairness evaluation s is simply XOR of the model’s prediction  and the auditor’s prediction , so given  and , one may back out . Also, more simply, we could have simply asked  from the auditor, which is still a binary question in this case. Then, given a dataset (x_i, y_i, f(x_i))_{i=1}^{n} where n is the total number of feedbacks we have gotten from the auditor, we can once again measure the empirical unfairness (e.g. false positive rate difference) with respect to f(x) from this dataset $(x_i, y_i, f(x_i))_{i=1}^{n}$. The convergence rate of unfairness estimated this way should be  as before.
>
> **Response 3**
> We agree that in case of binary classification the auditor's true outcome can be extracted by performing XOR operation. However, the intrinsic evaluation f adopted by the auditor is usually a subconscious phenomenon and the auditor may find it impossible to formally state his/her intrinsic rule explicitly. For instance, consider Adult income dataset where the task is to predict income of an individual over the intervals >$10,000, >$20,000, >$30,000, and >$40,000. In a such scenario, it is difficult for a human auditor to precisely classify the individual but identifies fairness violations. Similar idea was implemented by Stephen Gillen et al. in "Online Learning with an Unknown Fairness Metric", where the oracle/auditor identifies fairness violations, but does not quantify their extent. In other words, the auditor *recognizes unfairness when he/she sees it*, but cannot enunciate a quantitative fairness metric over individuals. Although our approach works for systems beyond binary classifiers, we could not present validation results for such systems, since there is no open real-world dataset for non-binary classifiers in the context of algorithmic fairness.
>
> **Weakness 4**
> The proposed method of this paper is to learn $\hat{f}$ from this dataset (x_i, y_i, f(x_i))_{i=1}^{n} and measure the unfairness on a separate dataset for which we don’t have the auditor’s feedback by imputing the auditor’s feedback with $\hat{f}$. Information theoretically speaking, measuring unfairness with respect to $\hat{f}$ on this bigger dataset cannot be better than the unfairness measure with respect to the original (x_i, y_i, f(x_i))_{i=1}^{n}. Of course, collecting f(x) (i.e. asking the auditors for their opinion) is an expensive process, but there's no free lunch in that estimating f with some hat{f} and measuring the unfairness simply with respect to $\hat{f}$ is not a viable alternative as it will further propagate the error of hat{f}.
>
> **Response 4**
> We concur with the reviewer regarding the potential unfairness that may arise due to the learning error. We plan to address this concern by capturing the error between the true non-comparative fairness $\mathbb{E}_x[s]$ and the predicted non-comparative fairness $\mathbb{E}_x[\hat{s}]$ of the auditor, where $\hat{s}$ is the predicted feedback based on the estimated auditor's LAM model. Unfortunately, due to travel disruptions this summer, we were unable to complete this task. We request the reviewer to provide us this additional two weeks so that we can respond diligently to this concern.
>
> **Weakness 5**
> Section 4.2 shows what one can show if the auditor’s intrinsic rule f is eps-LAM with respect to the model g (i.e. f makes a similar prediction as the model g). More formally, they show that if f and g is similar in the sense of eps-LAM, g’s fairness with respect to some fairness notion implies f’s fairness and vice versa. As discussed above, I don’t immediately see how auditing for unfairness of f and/or g via this relationship between them is more helpful as compared to measuring the unfairness of f and/or g directly. Information theoretically speaking, estimating unfairness of f or g directly can only be better than measuring indirectly via this relationship I believe.
>
> **Response 5**
> In order to compute the unfairness of the auditor $f$, we need to consider the system's output labels $g(x)$ as true labels. However, as discussed earlier, we rarely have access to true labels in fairness-related applications. For example, let's consider a classifier that decides to accept or reject loan applications. Historically, many banking firms have practised redlining by intentionally rejecting loans to qualified Black applicants. This leads to the problem of \emph{label bias}. Based on ground truth, loans should have been given to these applicants, but they were purposefully designated as defaulters. The classifiers trained on such mislabeled data can mistakenly reject a loan.

---

> ### Author Response · Authors · 2022-08-09
> **Our Responses to Reviewer Syg5 (Part 3)**
>
> **Weakness 6**
> Also, there’s some discussion about learning the auditor’s prediction hat{f} from the elicited feedback from the auditor in section 4.3. This learned hat{f} is only useful if (1) the learned intrinsic rule hat{f} is actually close to f and (2) f is actually eps-LAM to g. However, whether or not the auditor’s intrinsic rule f is eps-LAM to g is not something that we can control, so I’m not sure exactly what theorem 1 is trying to convey.
>
> **Response 6**
> We agree with reviewer's concern. Consider the following proof. If the auditor twin $\hat{f}$ satisfies $\epsilon$-LAM with respect to the system $g$, then $d(\hat{f}(x), g(x)) < \epsilon$ for all $x \in \mathcal{X}$. Since unfairness in $f$ can only be better than $\hat{f}$, we can infer that the auditor $f$ also satisfies $\epsilon$-LAM with respect to $g$ as follows $d(f(x), g(x)) < \epsilon$ for all $x \in \mathcal{X}$. From the above two equations, we can say that $d(\hat{f}(x), f(x)) < 2\epsilon$.
>
> **Weakness 7**
> Furthermore, g is a blackbox ML model we have access to so I don’t understand why there is a need to learn hat{g} at all. In addition, in theorem 1, it’s stated that it outputs hat{f} and hat{g} such that d(f(x), g(x)) <= epsilon with high probability. As explained before, note that whether or not auditor’s intrinsic rule f is eps-LAM with respect to the model g (how close f is to g) does not depend on the algorithm used to learn hat{f} and hat{g} and the number of samples we have to learn hat{f} and hat{g}. So, it cannot be that the algorithm always outputs $d(f(x), g(x)) <= eps$ with high probability always. So, I take it that the intended theorem statement is that $d(hat{f}(x), hat{g(x)})<=epsilon$. However, if the intrinsic rule f is very different from the ML model g (it is not eps-LAM), then I think it’s impossible to have both $d(hat{f}(x), hat{g(x)})<=epsilon$ and hat{f} is close to f.
>
> **Response 7**
> We agree with reviewer's comment. However, while evaluating real-world ML systems such as COMPAS, we do not have access to the original data used to train the model. In other words, we cannot query and obtain $g(x)$ for some new $x \in \mathcal{X}$. The only way the ML algorithm's label can be found using another machine learning model which mimics the original system. This necessitates us to consider the error in $\hat{g}$ as well. We explain this in detail in our proof for Theorem 1.
>
> **Weakness 8**
> Finally, PAC learnability is always stated with respect to some hypothesis class regardless of whether we are in a realizable setting or agnostic setting. It seems like there’s an inherent assumption that e are in a realizable setting for the theory part.
>
> **Response 8**
> We agree with the reviewer's comment. Our result in Theorem 1 corresponds to PAC learnability in a realizable setting. We clarified this in our paper in the definition of PAC learnability and in Theorem 1.
>
> **Weakness 9**
> In terms of experiment, my understanding is that they have difference heuristics rules $f$ for each dataset that the auditor may possibly use (as defined in page 10) and try to learn this rule f via different methods (logistic regression, SVM, and random forest) from some training dataset \left\{x_i, y_i, f(x_i)\right\}_{i=1}^{n}and measure the unfairness of this learned rule hat{f} on a bigger dataset. However, as I voiced the concern above, one needs to compare this to direct unfairness measurement from \left\{x_i, y_i, f(x_i)\right\}_{i=1}^{n}.
>
> **Response 9**
> As stated earlier, we are working on this direct unfairness measurement $E_x(s)$ and comparing it with $E_x(\hat{s})$ empirically. However, we request an additional two weeks in order to complete this work diligently.
>
> **Weakness 10**
> Finally, there needs to be more explanation about individual fairness definition used here. The individual definition proposed in this paper is slightly different than the original individual fairness definition which is $|f(x) - f(x')| <= d(x,x') + slack$ where f is some soft classifier (f outputs some number from [0,1] instead of \{0,1\}).
>
> **Response 10**
> As suggested, we explained the need to consider a more generalized individual fairness notion for this paper.

---

> ### Author Response · Authors · 2022-08-09
> **Our Responses to Reviewer Syg5 (Part 4)**
>
> **Requested Changes**
>
> More than anything, the paper needs to explain the fundamental benefit of using this LAM framework as opposed to directly measuring the unfairness of f and/or g directly.
>
> -- *Done*.
>
> For the experiment, it is insufficient to simply report the unfairness measured by imputing the auditor’s feedback on some bigger dataset with the learned hat{f}. (1) f’s true unfairness: Because we actually have the intrinsic rule f, f’s unfairness measured on this bigger dataset should be reported. (2) f’s empirical unfairness: the epirical unfairness measured on the original training dataset \left\{x_i, y_i, f(x_i)\right\}_{i=1}^{n} should be reported.
>
> -- *In progress. Requesting two weeks of additional time.*
>
> It needs to be shown that the unfairness measured by imputing the auditor’s feedback on some bigger dataset with learned hat{f} is closer to (1) f’s true unfairness than (2) f’s empirical unfairness is to (1) in order to establish that LAM is better than directly measuring the unfairness of f and/or g directly.
>
> -- *In progress. Requesting two weeks of additional time.*

---

> ### Author Response · Authors · 2022-09-13
> **Our updated responses for Reviewer Syg5**
>
> **Weakness 4**
> The proposed method of this paper is to learn $\hat{f}$ from this dataset (x_i, y_i, f(x_i))_{i=1}^{n} and measure the unfairness on a separate dataset for which we don’t have the auditor’s feedback by imputing the auditor’s feedback with $\hat{f}$. Information theoretically speaking, measuring unfairness with respect to $\hat{f}$ on this bigger dataset cannot be better than the unfairness measure with respect to the original (x_i, y_i, f(x_i))_{i=1}^{n}. Of course, collecting $f(x)$ (i.e. asking the auditors for their opinion) is an expensive process, but there's no free lunch in that estimating $f$ with some $\hat{f}$ and measuring the unfairness simply with respect to $\hat{f}$ is not a viable alternative as it will further propagate the error of $\hat{f}$ (i.e. how different $\hat{f}$ is from $f$).
>
> **Response 4**
> We concur with the reviewer regarding the potential unfairness that may arise due to the learning error. In our paper, we provide theoretical (Section 4, Theorem 2) and practical guarantees (Sections 8.1 and 8.2) emphasizing that the errors in estimating the auditor's intrinsic rule $f$ and the black-box classifier $g$ have to be minimum to achieve a better estimate of auditor twin's LAM. Specifically, given a small set of data samples, we practically show that the auditor's subjective labels can be learned with high accuracy while maintaining measure of unfairness between $f$ and $\hat{f}$ to a minimum.
>
> **Weakness 8**
> Finally, PAC learnability is always stated with respect to some hypothesis class regardless of whether we are in a realizable setting or agnostic setting. It seems like there’s an inherent assumption that e are in a realizable setting for the theory part.
>
> **Response 8**
> We agree with the reviewer's comment. Our result in Theorem 1 corresponds to PAC learnability in both realizable and non-realizable settings. We clarified this in our paper in the definition of PAC learnability and in Theorem 1.
>
> **Weakness 9**
> In terms of experiment, my understanding is that they have difference heuristics rules $f$ for each dataset that the auditor may possibly use (as defined in page 10) and try to learn this rule f via different methods (logistic regression, SVM, and random forest) from some training dataset {x_i, y_i, f(x_i)}_{i=1}^{n} and measure the unfairness of this learned rule hat{f} on a bigger dataset. However, as I voiced the concern above, one needs to compare this to direct unfairness measurement from $\left\{x_i, y_i, f(x_i)\right\}_{i=1}^{n}$.
>
> **Response 9**
> In Section 8, we practically show that, if the simulated auditor's intrinsic rule is learned with minimal error, the absolute error between the true fairness measure $\mathbb{E}_x[s]$ and the empirical fairness measure $\mathbb{E}_x[\hat{s}]$ is close to zero. In addition, using real human feedback data, we show that the majority of crowd workers' intrinsic rule can be learned with minimal error.
>
> **Requested Changes**
>
> More than anything, the paper needs to explain the fundamental benefit of using this LAM framework as opposed to directly measuring the unfairness of f and/or g directly. -- Done.
>
> For the experiment, it is insufficient to simply report the unfairness measured by imputing the auditor’s feedback on some bigger dataset with the learned hat{f}. (1) f’s true unfairness: Because we actually have the intrinsic rule f, f’s unfairness measured on this bigger dataset should be reported. (2) f’s empirical unfairness: the epirical unfairness measured on the original training dataset $\left\{x_i, y_i, f(x_i)\right\}_{i=1}^{n}$ should be reported.
>
> *-- Done*
>
> It needs to be shown that the unfairness measured by imputing the auditor’s feedback on some bigger dataset with learned hat{f} is closer to (1) f’s true unfairness than (2) f’s empirical unfairness is to (1) in order to establish that LAM is better than directly measuring the unfairness of f and/or g directly.
>
> *-- Done*

---

### Review · Reviewer_p9qv · 2022-07-25

**Summary Of Contributions:**

This paper attempts to put fairness definitions into (simulated and crowd-worked) practice for auditing purposes. The idea is compare auditor decisions to automated ones in a formal way, in order to validate the fairness/unfairness of decisions (by attempting to model human auditor decisions that include feedback, and providing a kind of reputation score for each auditor). In particular, they propose a new model (LAM), which helps judge fairness/unfairness of decisions (predicated on the assumption that auditors compare their own decision outcomes to the automated rule provided). The authors then use this model to show hat fairness notions can be guaranteed (when the auditor themselves satisfy the respective fairness notion). The authors prove a lower bound for PAC learning estimations of LAM. Lastly, the authors perform experiments on commonly-used datasets in algorithmic fairness to evaluate their results. They also rely on a previously-collected human auditing dataset to complement their simulations.

**Broader Impact Concerns:**

I have some concerns about how auditors and reputation are modeled. I am not sure 96% accuracy on modeling auditors is sufficient for high-stakes decisions to claim reliability of the methods presented in practice. I am also not convinced that within-auditor variation is considered sufficiently as a possible problem (and, I would hazard to guess, is partly a problem with the accuracy of the models). I discuss this above concerning the popularized terms "pattern noise" and "level noise". I am also concerned about the flattening of expertise in the aggregated reputation matrix. While it may seem sound mathematically, in my opinion it devalues domain expertise and context (and is suggestive that devaluing such expertise is okay, in the strictest read), and I worry that such abstraction creates additional risks.

**Requested Changes:**

* Please see points in Weakness and Nits concerning clarity and typo issues (some of which are correctness problems, in my opinion)

* Further discussion / justification of why one aggregated reputation matrix makes sense, especially since (in my opinion) a natural interpretation of reputation hinges on domain expertise and some notion of problem-dependence (see weaknesses above). That is, it seems unlikely that an auditor who is an expert in one domain would similarly be an expert in a completely different domain.

* Justification of 96% accuracy on modeling auditors (see weaknesses above)

* Justification of discretization in the \epsilon-LAM definition (see weaknesses)

**Strengths And Weaknesses:**

Strengths:

* This paper attempts to provide a solution to a very tricky problem -- a concrete, measurable notion of auditor fairness that is compatible / comparable with algorithmic fairness, and a relationship between automated and manual metrics that can be defined rigorously/formally to provide guarantees about overall fairness of automated decisions that involve auditing/ feedback.

* The paper is ambitious in its scope. In addition to the above, it also tries to come up with an understandable, rich way of assessing auditor quality. However, I believe there are some issues with the current formulation (see below).

* The paper does not limit itself to one fairness notion, but rather covers most (arguably all) of the most-commonly-used fairness definitions.

Weaknesses:

* While the authors take care to mention cases in which context can matter (e.g., for choice of fairness metric), the notion of reputation does not capture this. First, as noted by the authors on page 11, reputation treats all fairness measures equally (when in fact, for example, there may be a clear argument not to use statistical parity in some cases, which the authors themselves indicate on p. 3). Second, the decision context *also* matters, I imagine, in terms of auditor expertise/ ability to audit (concretely, why would an auditor on COMPAS necessarily perform similarly on Adult? Would it not be more useful and prudent to develop reputation scores on a per-domain basis, rather than as one aggregate number?)

* The writing is often unclear. Please see low-level examples below. In general, the paper needs a close read for typos and grammatical edits; this is not just about aesthetics, as I think in some places it impacts correctness (again, see examples listed below for smaller things). For one of the more important examples: \epsilon-LAM does not actually define semantically what s=1 and s=0 corresponds to in words. I could guess this based on the math in (6), but being explicit here would be very helpful. Additionally, it seems like \epsilon-LAM is a discretized measure of real-number agreement, but then in later proofs / propositions it seems to be talked about colloquially/informally as if it is a statement on fairness. For example, on page 6 / in proposition 1: "However, note that if g is \epsilon-LAM with respect to f, then d(g(x_i), f(x_i)) < \epsilon, which seems to be talking about s=0 specifically (not the fact that \epsilon-LAM applies as a model, which would include the applicability of s=1 as a possibility). Do the authors mean e-LAM-agreement here and elsewhere when this type of phrasing is used?

* While I understand why the authors treat s as an integer / use \epsilon-LAM to discretize continuous values, it does seem like there is (potentially) interesting or useful information being lost concerning the extent to which agreement/disagreement between an auditor and the system occurs. Even if the authors do not want to explore that case, it would be worth calling this out as something recognized / relegated to future work or out of scope. Otherwise, it seems like a potential weakness of the problem formulation (when it could be a justified choice).

* I am not sure that 96% accuracy (concerning modeling auditors) is accurate enough to claim fidelity in high-stakes decisionmaking contexts (like those that algorithmic fairness, and thus this paper, contends with) (p. 13). There is a lot of work on just how noisy/ non-statistical human decisionmaking is in such context. See, for example, work by Cass Sunstein, Daniel Kahneman concerning judge rulings in the US, to illustrate this point. Level errors concern differences between judges on the same decision (e.g., some may be more lenient than others) -- this seems to be the kind of bias the authors account for. In contrast, pattern errors can show relative deviations within a particular judge to reveal seemingly inexplicable internal inconsistencies (e.g., across judges, we can compute the deviation for a particular decision, and see if a particular judge is harsher or more lenient than their personal average in some cases). Pattern errors can come about for all types of reasons (waking up on the wrong side of the bed before heading to court, for example), which are not the same as bias but can nevertheless lead to unfairness. This type of error is notoriously difficult to model, and I imagine is one of the sources of error in the 4% error (though would need to have data that could show this -- that a particular auditor varied outcomes on two close-to-identical or identical cases). This makes me suspect that doing much better than the authors have done is likely hard (if not intractable), but I'm not sure that 96% is reliable enough, practically speaking

Nits and low-level feedback:

* There are a few sentences that do not parse. For example, the second to last sentence in 1.1 is a bit jumbled / took a few reads to grok. As another example, the last sentence in the section on Statistical Parity on p.3 is missing a clause. In general, I think the paper could do with a close read for typos and grammatical edits. I also believe there is a typo in (11); should the first plus sign on the right-hand side of the equation be a minus sign?

* It would be helpful in definitions to consistently define symbols and types (e.g. integer, real number). For example, kappa and delta in 2.2 should be given types; d and \mathcal{D} should both be defined before used. As another example, the circle plus / XOR in Remark 1 should be defined, as this symbol is often overloaded in math as some kind of addition operator (sometimes direct sum, sometimes something else). I wasn't sure what it meant until I got to the end of 2 paragraphs later (on p. 5), where it is described as XOR. In (17), is F female and M male?

* Error bars on Figure 4? Or was just one model run and I'm misunderstanding something?

---

> ### Author Response · Authors · 2022-08-09
> **Our Responses to Reviewer p9qv (Part 1)**
>
> *We sincerely thank the reviewer for valuable feedback. However, the feedback has proved that our work needs major changes. Unfortunately, we were unable complete all the major changed in the given time. We request the reviewer to give two-week time to complete all the concerns and submit the revised version.*
>
> Our response to Reviewer p9qv's comments and the corresponding changes in the original paper will be highlighted in CYAN color.
>
> **Strengths**
>
> This paper attempts to provide a solution to a very tricky problem -- a concrete, measurable notion of auditor fairness that is compatible / comparable with algorithmic fairness, and a relationship between automated and manual metrics that can be defined rigorously/formally to provide guarantees about overall fairness of automated decisions that involve auditing/ feedback.
>
> The paper is ambitious in its scope. In addition to the above, it also tries to come up with an understandable, rich way of assessing auditor quality. However, I believe there are some issues with the current formulation (see below).
>
> The paper does not limit itself to one fairness notion, but rather covers most (arguably all) of the most-commonly-used fairness definitions.
>
> **Response to Strengths**
> We thank the reviewer for the feedback and sincerely appreciate it.
>
> **Weakness 1**
> While the authors take care to mention cases in which context can matter (e.g., for choice of fairness metric), the notion of reputation does not capture this. First, as noted by the authors on page 11, reputation treats all fairness measures equally (when in fact, for example, there may be a clear argument not to use statistical parity in some cases, which the authors themselves indicate on p. 3).
>
> **Response 1**
> As suggested, we are investigating different potential reputation metrics (e.g. $||R||_F$, $\text{Tr}(RR^T)$, and other possibilities). Unfortunately, due to travel disruptions this summer and just two weeks of response time, we were unable to complete this task. We request the reviewer to provide us this additional two weeks so that we can respond diligently to this concern.
>
> **Weakness 2**
> Second, the decision context also matters, I imagine, in terms of auditor expertise/ ability to audit (concretely, why would an auditor on COMPAS necessarily perform similarly on Adult? Would it not be more useful and prudent to develop reputation scores on a per-domain basis, rather than as one aggregate number?)
>
> **Response 2**
> We completely agree with the reviewer. In fact, the main idea behind considering non-comparative fairness perspective is to account for contextual elements within the fairness evaluation process. This is evident from our experimental results as well, since the reputation score computed is only for COMPAS dataset. We swapped Section 4 and 5 to resolve this confusion.
>
> **Weakness 3**
> The writing is often unclear. Please see low-level examples below. In general, the paper needs a close read for typos and grammatical edits; this is not just about aesthetics, as I think in some places it impacts correctness (again, see examples listed below for smaller things). For one of the more important examples: $\epsilon$-LAM does not actually define semantically what s=1 and s=0 corresponds to in words. I could guess this based on the math in (6), but being explicit here would be very helpful.
>
> **Response 3**
> We thank the reviewer for the suggestion. In our proposed LAM, if the distance between the system's output label $g(x)$ and the subjective evaluation of the auditor $f(x)$ is greater than or equal to some $\epsilon$, then the auditor with deem $g(x)$ as unfair. We added the semantic meaning of the score $s$ in Section 3 paragraph 2.

---

> ### Author Response · Authors · 2022-08-09
> **Our Responses to Reviewer p9qv (Part 2)**
>
> **Weakness 4**
> Additionally, it seems like $\epsilon$-LAM is a discretized measure of real-number agreement, but then in later proofs / propositions it seems to be talked about colloquially/informally as if it is a statement on fairness. For example, on page 6 / in proposition 1: "However, note that if g is $\epsilon$-LAM with respect to f, then $d(g(x_i), f(x_i)) < \epsilon$, which seems to be talking about s=0 specifically (not the fact that $\epsilon$-LAM applies as a model, which would include the applicability of s=1 as a possibility). Do the authors mean e-LAM-agreement here and elsewhere when this type of phrasing is used?
>
> **Response 4**
> We agree with the reviewer. In order to alleviate this confusion, we rephrased our $\epsilon$-LAM definition as follows:
> Definition[$\epsilon$-Latent Assessment Model]
> An auditor is said to satisfy $\epsilon$-LAM if there exists a tuple $(\mathcal{X}, \mathcal{Y}, d, f, \epsilon)$ such that the auditor compares the system's output $g(x)$ with an intrinsic judgment $f(x)$ using a distance metric $d$ in following manner
> \begin{equation}
> d \big( g(x), f(x) \big) < \epsilon,
> \end{equation}
> and reveal his/her binary feedback as
> \begin{equation}
> s\big( x, y=g(x), \hat{y}=f(x) \big) = \begin{cases}
> 1, & \text{if } \ d \big( g(x), f(x) \big) \geq \epsilon,
> \\\\[1ex]
> 0, & \text{otherwise.}
> \end{cases}
> \end{equation}
>
> **Weakness 5**
> While I understand why the authors treat s as an integer / use $\epsilon$-LAM to discretize continuous values, it does seem like there is (potentially) interesting or useful information being lost concerning the extent to which agreement/disagreement between an auditor and the system occurs. Even if the authors do not want to explore that case, it would be worth calling this out as something recognized / relegated to future work or out of scope. Otherwise, it seems like a potential weakness of the problem formulation (when it could be a justified choice).
>
> **Response 5**
> We request two weeks of additional time to properly answer this concern.
>
> **Weakness 6**
> I am not sure that 96\% accuracy (concerning modeling auditors) is accurate enough to claim fidelity in high-stakes decisionmaking contexts (like those that algorithmic fairness, and thus this paper, contends with) (p. 13). There is a lot of work on just how noisy/ non-statistical human decisionmaking is in such context. See, for example, work by Cass Sunstein, Daniel Kahneman concerning judge rulings in the US, to illustrate this point. Level errors concern differences between judges on the same decision (e.g., some may be more lenient than others) -- this seems to be the kind of bias the authors account for. In contrast, pattern errors can show relative deviations within a particular judge to reveal seemingly inexplicable internal inconsistencies (e.g., across judges, we can compute the deviation for a particular decision, and see if a particular judge is harsher or more lenient than their personal average in some cases). Pattern errors can come about for all types of reasons (waking up on the wrong side of the bed before heading to court, for example), which are not the same as bias but can nevertheless lead to unfairness. This type of error is notoriously difficult to model, and I imagine is one of the sources of error in the 4\% error (though would need to have data that could show this -- that a particular auditor varied outcomes on two close-to-identical or identical cases). This makes me suspect that doing much better than the authors have done is likely hard (if not intractable), but I'm not sure that 96\% is reliable enough, practically speaking
>
> **Response 6**
> The behavioral decision models discussed by the reviewer can all be accomodated within the intrinsic evaluation $f$ in our proposed LAM. However, in this paper, we do not describe the specifics of how $f$ is manifested in human decision making, but rather adopt a model-agnostic way to characterize human behavior using machine learning. We agree that the error in mimicing auditor feedback can potentially raise flags from a fairness perspective in the comprehensive evaluation of the high-dimensional system in practice. However, this is the first attempt towards addressing this concern, and we hope to improve our learning performance in the future. In the worst case, if reducing this error becomes intractable, we will adopt a model-based learning approach as suggested by the reviewer in our future work. We discuss this in Section 9 of the paper.

---

> ### Author Response · Authors · 2022-08-09
> **Our Responses to Reviewer p9qv (Part 3)**
>
> **Nits and low-level feedback**
>
> There are a few sentences that do not parse. For example, the second to last sentence in 1.1 is a bit jumbled / took a few reads to grok. As another example, the last sentence in the section on Statistical Parity on p.3 is missing a clause. In general, I think the paper could do with a close read for typos and grammatical edits. I also believe there is a typo in (11); should the first plus sign on the right-hand side of the equation be a minus sign?
>
> *-- Fixed.*
>
> It would be helpful in definitions to consistently define symbols and types (e.g. integer, real number). For example, kappa and delta in 2.2 should be given types; d and $\mathcal{D}$ should both be defined before used. As another example, the circle plus / XOR in Remark 1 should be defined, as this symbol is often overloaded in math as some kind of addition operator (sometimes direct sum, sometimes something else). I wasn't sure what it meant until I got to the end of 2 paragraphs later (on p. 5), where it is described as XOR. In (17), is F female and M male?
>
> *-- Fixed.*
>
> Error bars on Figure 4? Or was just one model run and I'm misunderstanding something?
>
> *-- The bars represent the number of crowd worker/auditors whose responses are learned and predicted using three different frameworks: Logistic regression, decision tree and SVM.*
>
> **Requested Changes**
>
> Please see points in Weakness and Nits concerning clarity and typo issues (some of which are correctness problems, in my opinion)
>
> *-- Fixed*
>
> Further discussion / justification of why one aggregated reputation matrix makes sense, especially since (in my opinion) a natural interpretation of reputation hinges on domain expertise and some notion of problem-dependence (see weaknesses above). That is, it seems unlikely that an auditor who is an expert in one domain would similarly be an expert in a completely different domain.
>
> *-- Addressed in responses.*
>
> Justification of 96\% accuracy on modeling auditors (see weaknesses above)
>
> *-- Addressed in responses.*
>
> Justification of discretization in the $\epsilon$-LAM definition (see weaknesses)
>
> *-- Addressed in responses.*

---

> ### Author Response · Authors · 2022-09-13
> **Our updated response for Reviewer p9qv**
>
> **Weakness 1**
>
> While the authors take care to mention cases in which context can matter (e.g., for choice of fairness metric), the notion of reputation does not capture this. First, as noted by the authors on page 11, reputation treats all fairness measures equally (when in fact, for example, there may be a clear argument not to use statistical parity in some cases, which the authors themselves indicate on p. 3).
>
> **Response 1**
>
> As suggested, we compared our proposed reputation mechanism with other scoring functions on real human feedback data in Section 8.3. Specifically, we compare the Frobenius norm reputation of the auditor with two different scoring functions i.e., eigenvalues and spectral norm of the reputation matrix $R$. Practical results show a monotonic relationship between Frobenius norm based reputation score and eigenvalue/spectral-norm based scoring mechanisms.
>
> **Nits and low-level feedback**
>
> Error bars on Figure 4? Or was just one model run and I'm misunderstanding something? -- *Figure is updated for more insights.*

---

### Decision · Action_Editors · 2022-09-20

**Recommendation:** Reject

**Comment:**

## Paper overview

The paper introduces the latent assessment model (LAM) framework for performing "non-comparative" fairness assessments.  They also connect between their epsilon-LAM definitions and comparative (e.g., group and individual notions of fairness), introduce a reputation measure to assess auditor bias simultaneously across different fairness metrics, and conduct experiments on COMPAS, Adult, and German Credit data.

## Decision criteria

The criteria for acceptance at TMLR is twofold:

* Are the claims made in the submission supported by accurate, convincing and clear evidence?
* Would at least some individuals in TMLR's audience be interested in knowing the findings of this paper?

## Decision Recommendation

Having carefully reviewed the latest version of the manuscript, reviewer feedback, and author responses, I recommend this submission for **rejection**.  I agree with the reviewers that, even after revision and author clarification, the approach remains poorly-motivated.  I share in the reviewers' common concern that the authors have not made a compelling case for when their approach would be more appropriate (or even equally appropriate) to other forms of fairness assessment.  I also agree that the paper is shaky on its fundamentals: the authors cast their work as measuring fairness, but what the approach directly measures is disagreement (and that, only through 'digital twin' modeling).  The connection between fairness and disagreement is assumed throughout the work (e.g., if an auditor is fair, and a model agrees with the auditor, then the model will be fair, and vice versa).  Since the authors seek to motivate the work at the outset in settings where experts fail to provide reliable annotations, the proposed solution of examining (actual or inferred) disagreement with human auditors is on shaky footing from the outset.  The submission in my view falls short on both acceptance criteria: (i) the work is not well-motivated from the outset, and thus does not produce convincing fairness claims in the end; and (ii) I expect that general TMLR readership would have the same concerns and thus generally be uninterested in the methods and experimental results.

## Comments

### Motivation

All reviewers have identified disconnects between the proposed approach and the stated motivation.  I will summarize a few of my own as well.  First, I do not see how the work is motivated in the criminal justice, consumer lending, or insurance settings, or other settings beyond these that the authors mention.  In these settings, models are generally trained on historical data to predict a particular observed target.  In the criminal justice context this is recidivism; in lending it's default; in life insurance it's lifetime.  (I note that is contradictory to statements made in Response 2 of in Part 1 of response to Syg5, where the authors claim that true labels are inaccessible in social applications.)

Even if we were to consider a setting where models are trained on subjectively labelled data (e.g., on, say, employee performance evaluations) it's hard to see how comparing to auditor assessments then tells us anything about fairness.  If we can't trust subjective assessments, why would disagreement between a model's predictions and (even directly observed) auditor assessments tell us anything?  In those cases wouldn't we just want to learn the auditor's model and use that as our ML system at the end of the day anyway?

At a high level, the premise of the paper is that if we have system A and system B, and system B has desirable properties---which the authors call fairness, but which I do not see as being specific to fairness in any way---then if A and B agree often enough we may be able to conclude that A also has desirable properties. As I note below, if A and B disagree, we cannot conclude that B does not have desirable properties.

The authors should reflect on when disagreement between a model and human auditor (or two models, even) is an indication of anything.  Due to predictive multiplicity (e.g., see Marx 2020, Chouldechova 2017) one can have settings where two models are equally accurate and yet disagree on a large number of their classifications.  So, likewise, we can have a model and an auditor who are equally accurate and yet frequently disagree.  So this tells us that we can't use disagreement to meaningfully assess model accuracy in many cases.  How about fairness?  Here the authors have not appropriately distinguished between fairness and accuracy.  What does fairness on a given instance mean if not correctness with respect to some merit (i.e., target $Y$, which maybe is hard to observe)?

In general, discussions of motivating examples use terminology or consider settings that are disconnected from their respective domains.   E.g., at a point the authors write "a parole application by a Black applicant is to deny the bail".  Bail pertains to pre-trial decision-making.  Parole is a post-conviction decision.  Later the authors write "A banking system would evaluate both the applications via collecting
information such as gender, race, address, credit history, collateral, and his/her ability to pay back."  Given that e.g., the Equal Credit Opportunity ACT (ECOA) prohibits creditors from inquiring about race and sex (where it is not by some exception required to do so), the setting described is once again far removed from reality.

In certain places the authors aim to assess the fairness of the auditor rather than the system, generally by assuming that it's the system that's fair in those instances.  While I agree that given a fair system A one can compare various forms of feedback/assessments of system B to (maybe) determine if system B is also fair, why this is the path one would take is unclear.  One could even get the completely wrong conclusion from this type of approach compared to a more direct approach for something like statistical parity.  E.g., in the realm of predictive multiplicity two classifiers (be they human or machine) may disagree on a large number of classifications.  It is further possible for classifiers A and B to both satisfy statistical parity while disagreeing at high rates.  So for comparative notions of fairness, we would not use disagreement between two systems to indicate unfairness.

Overall, the work does not make particularly novel theoretical or mathematical contributions, so its merit rests largely on its utility and applicability.  In revising the work I encourage the authors to focus on producing at least one truly compelling real world example in which their approach would be preferred to (or at least equally preferable to) an existing strategy.

I agree with Reviewer p9qv

### Theoretical results

As Syg5 noted, the statement of Theorem 1 remains unclear. The statement couldn't possibly apply to any rules $f$ and $g$.  E.g., if $d(g,f) > \epsilon$ there wouldn't exist an $N$ guaranteeing the empirical $\epsilon$-LAM condition.  What is this theorem intending to establish?

### Additional references

A Chouldechova, M G'Sell "Fairer and more accurate, but for whom?" FAT/ML 2017.

Marx, Charles, Flavio Calmon, and Berk Ustun. "Predictive multiplicity in classification." International Conference on Machine Learning. PMLR, 2020.